# The collective burst mechanism of angular jumps in liquid water

Adu Offei-Danso[1,2], Uriel N. Morzan[1], Alex Rodriguez[1,3], Ali Hassanali [1] & Asja Jelic [1] ✉

Understanding the microscopic origins of collective reorientational motions in aqueous systems requires techniques that allow us to reach beyond our chemical imagination. Herein, we elucidate a mechanism using a protocol that automatically detects abrupt motions in reorientational dynamics, showing that large angular jumps in liquid water involve highly cooperative orchestrated motions. Our automated detection of angular fluctuations, unravels a heterogeneity in the type of angular jumps occurring concertedly in the system. We show that large orientational motions require a highly collective dynamical process involving correlated motion of many water molecules in the hydrogen-bond network that form spatially connected clusters going beyond the local angular jump mechanism. This phenomenon is rooted in the collective fluctuations of the network topology which results in the creation of defects in waves on the THz timescale. The mechanism we propose involves a cascade of hydrogen-bond fluctuations underlying angular jumps and provides new insights into the current localized picture of angular jumps, and its wide use in the interpretations of numerous spectroscopies as well in reorientational dynamics of water near biological and inorganic systems. The role of finite size effects, as well as of the chosen water model, on the collective reorientation is also elucidated.

Hydrogen-bond (H-bond) network fluctuations in water are at the heart of a wide range of physical, chemical, and biological processes, ranging from proton transfer in the ionization of water[1,2] to the folding of proteins and aggregation of molecules in solution[3,4]. Since water molecules are characterized by a rather large dipole moment, which in turn leads to directed interactions between them, the molecular reorientation mechanism has attracted significant interest[5–8].

Over a decade ago, Laage and Hynes demonstrated that water rotations do not occur solely via small diffusive steps, but instead typically involve sudden large-amplitude angular jumps[9–12]. The mechanism by which large and quick rotations of a water molecule happen is due to fluctuations in the local coordination patterns of neighboring waters suggesting a process that involves at least three water molecules which alters the local H-bond network topology.

When these jumps occur, do other water molecules in this network remain as spectators or are they active participants in a more collective process?

One of the enormous hurdles in determining a clear answer to this question originates from the difficulty in disentangling fluctuations occurring over a wide spread of both length and timescales, that create and form labile hydrogen bonds with a plethora of hydrogen-bond network patterns. On the other hand, numerous experiments ranging from dielectric spectroscopy[13] to time-dependent vibrational spectroscopy[6–8,14,15] indicate the presence of collective and concerted processes[7,8,16–18] underlying reorientational dynamics in water.

Here, using an automated detection of angular motions determined from molecular dynamics simulations of water, we illustrate a mechanism that unambiguously reveals the collective and correlated

[1]The Abdus Salam International Centre for Theoretical Physics, Trieste, Italy. [2]International School for Advanced Studies (SISSA), Trieste, Italy. [3]Dipartimento di Matematica e Geoscienze, Università degli Studi di Trieste, Trieste, Italy. ✉e-mail: asja.jelic@gmail.com

nature of water reorientation dynamics. By using an automatic protocol that identifies all abrupt angular motions, with no a priori assumptions on hydrogen-bonding, we demonstrate that there is a heterogeneity in the types of angular jumps that occur and that large reorientations are facilitated by a highly orchestrated motion of dozens of water molecules. We assert that these features are a generic property of the fluctuations in the local topology of the water H-bond network on the THz timescale. We also show that regions with lower local density serve as hot-spot sites in the network where large reorientations can occur simultaneously.

By conducting a finite-size analysis, as well as comparing with several water models, we also elucidate the generality of the collective reorientational mechanism. Specifically, while small-box simulations consisting of approximately 1000 waters, and commonly used in previous studies, tend to enhance the relative fluctuations of the H-bond network underlying the angular jumps, the extent of the spatially correlated clusters of the molecules involved, is not affected. Furthermore, the collective reorientational dynamics is found in several water models we examine, with subtle differences in the fraction of defects and asymmetries in the hydrogen-bonding structure both of which can affect the microscopic mechanisms.

## Results

In order to build our intuition on the collective nature of angular jumps, in Fig. 1a we highlight a specific event captured by our protocol illustrating all water molecules in the system that perform a large-amplitude angular reorientation within a selected time interval of approximately 350 fs in the course of the molecular dynamics (MD) simulation. From the total of 1019 water molecules, we find that around 5% of them undergo a major change in the direction of their dipole moment vector during this short time interval, suggesting that these are concurrent events. Many of these molecules are also located close by in space, forming apparent clusters of large jumping molecules. The round panels in the middle (Fig. 1b, c), show a close-up of several of these molecules before and after the angular jump, which is made evident by their initial and final dipole vector orientations. Shown in

the background are all the other water molecules in close vicinity to this event. For each reorienting molecule in the selected group, we quantified its angular fluctuation in Fig. 1d by following how its dipole vector changes in time through the time evolution of the angle it forms with respect to one of the axes of the laboratory coordinate system. While this only serves as a proxy for the angular fluctuations, it gives us a semi-quantitative measure of the size of the angular reorientation.

We see that the angular change of the dipole vectors of the eight molecules involved in this event, ranges between 60 and 120 degrees within the selected time interval. This type of angular change clearly modifies the direction in which the dipoles of these eight water molecules point to. Moreover, eye-balling the time series shown, indicates that these reorientation events are simultaneous and possibly correlated in space, which, as we will see shortly, requires a collective reorganization of the hydrogen-bond network. All this suggests that there is a highly coordinated dynamics underlying angular jumps, involving large number of molecules and diverse motions, which will be elucidated in the ensuing analysis.

Quantifying the angular jumps illustrated in Fig. 1d requires the identification of reaction coordinates that are highly non-local, involving several degrees of freedom that are challenging to identify by eye. With the aim to investigate the mechanisms behind the putative collective reorganization of water from the MD simulations, we developed an automatized protocol for identifying all the various angular fluctuations in water reorientation. Illustrations of the main steps of the procedure are shown in Fig. 2, while more details are given in Methods. For each water molecule in the system, the procedure works by spotting sudden changes along the trajectories of the vectors defined within the molecular frame of a water molecule, namely the dipole moment (DP) vector and the HH vector. As an outcome of the automatized protocol, for each detected angular change we obtain its starting time, duration ($\Delta t$), and angular magnitude ($\Delta\Theta$). As we will see later, the angular changes we identified, have a broad range of angular amplitudes and duration consistent with previous observations of the local angular jump picture. Therefore, we dubbed all detected molecular reorientations as angular swings, rather than angular jumps.

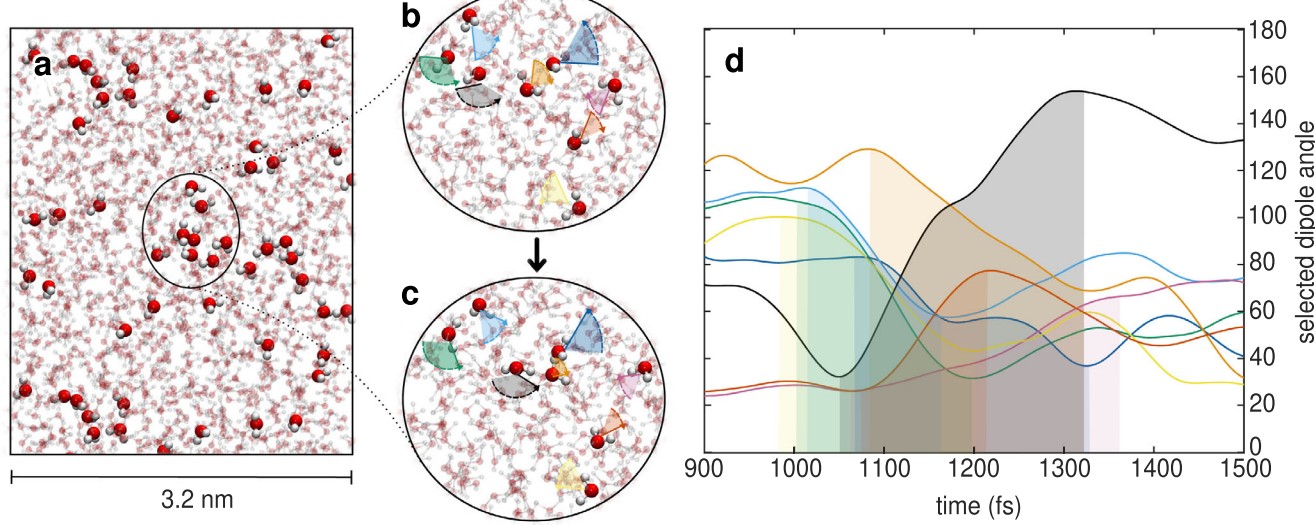

**Fig. 1 | Collective nature of angular jumps. a** Highlighted are all molecules undergoing angular reorientation of magnitude greater than 60 degrees in a box of 3.2 nm within the time interval of 350fs (which spans between time steps 1000fs and 1350fs in the MD simulation). They amount to around 5% of the total number of 1019 simulated molecules. **b, c** Close-ups of 8 of these molecules in a small region of the box at the start (**b**) and at the end (**c**) of a large angular jump as observed from the changes in their dipole vectors. The colored arcs outline the angular motion carried by the dipole vectors in the direction of the dashed arrow. Positions of the

molecules in **b, c** are slightly different due to translational motion during the observed time interval. **d** Change of the dipole vector in time for each of the selected molecules: We plot the time evolution of the angle it forms with respect to one of the axes of the laboratory coordinate system (for each molecule we show the component which changes most in this time interval). The regions between the start and the end of the angular jump are shaded by the colors of the corresponding molecules in panels **b, c**.

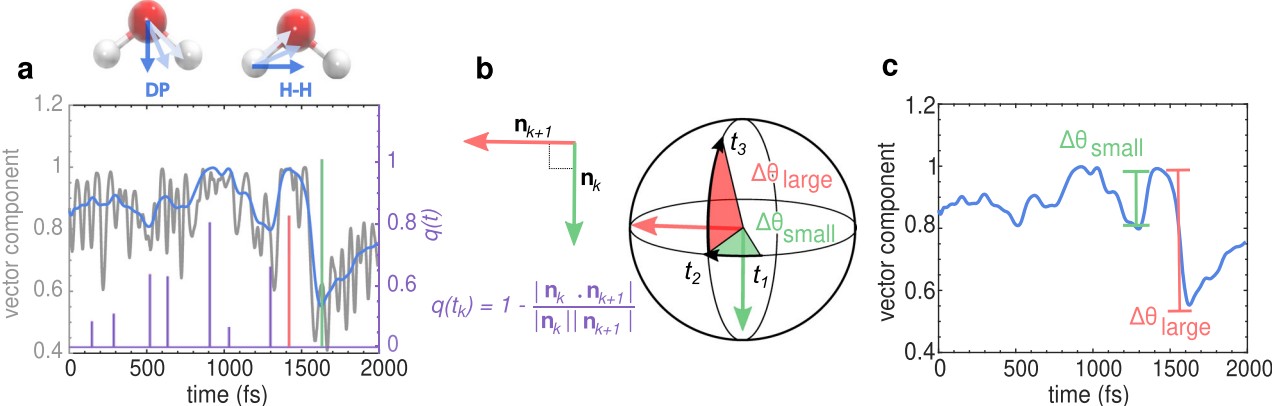

**Fig. 2 | Angular swing detection protocol. a** Definition of the dipole (DP) and HH vectors obtained for every molecule in the system and the plot showing the time series of the DP vector's x-component ($DP_x$) for a selected water molecule in a 2000 fs interval (gray line). The blue curve corresponds to the filtered time series eliminating the high frequency oscillations in the $DP_x$ vector. The purple line correspond to the function $q(t)$ defined in the Methods section that presents a spike at the beginning and the end of every angular swing. **b** Two successive swing events are detected by identifying the start and the end points of each angular swing. By employing function $q(t)$, we detect the instantaneous change in the direction of vector $\mathbf{n}(t)$ perpendicular to the plane of rotation of the DP (or HH) vector. This example shows our protocol for two successive swing events k-th and k+1-th in the time intervals ($t_1$, $t_2$) and ($t_2$, $t_3$), colored in green and red, respectively, detected from the filtered time series of the DP vector of one molecule. The direction of the vector normal to the plane of rotation is found to change only when transitioning from one interval to the other, i.e., from $\mathbf{n}_k$ (green arrow) during ($t_1$, $t_2$) time interval, to $\mathbf{n}_{k+1}$ (red arrow) during time interval ($t_2$, $t_3$). **c** Angular swings $\Delta\theta_{small}$ (small green swing) followed by $\Delta\theta_{large}$ (large red swing) are indicated on the filtered DP vector component time series. The start and end points correspond to extrema in the time series.

The latter is the predominant term in the literature, but it only refers to large-amplitude angular swings accompanied by H-bond breaking[9,10]. Here, we also analyze small angular fluctuations which would essentially correspond to small angular diffusive steps, that don't necessarily involve H-bond breaking.

## Jumps and defects come in waves

Underpinning the large angular jumps in the H-bond network, are fluctuations in the local topology of water molecules. Large angular motions usually create coordination defects which affect the hydrogen bonding patterns[10,19]. This, in turn, affects local topology of the H-bond network, leading to rearrangements of nearby water molecules and possibly further large reorientation events. Here, we examine the connection between changes in local topology and angular swings in more detail by quantifying the occurrence of these events in time for the entire ensemble of water molecules.

First, at every time step of the simulation, we calculate the number of waters in the system that are non-defective, i.e., those that accept two and donate two hydrogen bonds, and the number of all the other water molecules, which we refer to as defects. The time series of the fraction of defects with respect to the total number of water molecules in the system is shown in Fig. 3a over a time interval of 200 ps (see Supplementary Fig. 1 for the time evolution of the fraction of defects over a longer time scale). Interestingly, we observe that the fluctuations in the number of defective water molecules occur in waves. These oscillations reflect processes in the network which, on a picosecond timescale, for example, lead to the creation or annihilation of up to 10–20 defective water molecules in the network (large defect oscillations over short time scale are shown in Supplementary Fig. 1). Many of these bursts then appear to accumulate over a longer timescale leading to a slower process occurring on 10s of picoseconds that is evident in Fig. 3a.

Large changes in the number of defects in the network suggest that there is a collective process underlying the water reorientations in the H-bond network. This phenomenon is akin to similar effects proposing the role of defects in the mobility of water under supercooling[20]. To quantify better the collective nature of angular swings in the water network, we now look at how many angular swings occur simultaneously in the system. In similar spirit to the idea of

propagation of defects[13], we assume that the rearrangements of the local H-bond network due to a molecular reorientation can trigger further swings close by in time. Therefore, at every time step, we calculate how many molecules perform angular swings within a specific time interval around it.

Previous studies report that water reorientation occurs on a time scale of about 1ps that includes not only the angular jump itself, but also the breakage and forming of the H-bond before and after the jump[9,10]. Thus, we look at a time window of 1ps, within which we calculate all the angular swings happening in the system. By automatizing the detection of angular motions, we are able to identify all possible sudden angular changes in the waters' motion, spanning a wide range of duration and amplitude. However, large-amplitude swings are most likely the ones leading to water reorientation and change in local topology, as considered in the literature[9,10]. Moreover, the small-amplitude swings do not change much the orientation of the water molecule, and are less likely to affect local topology of the H-bond network.

In Fig. 3a, we plot the number of swings with the angular amplitude larger than 60 degrees, detected from the angular motion of both DP and HH vectors. We find that the number of concurrent large swings in the system fluctuates on the timescale of 10s of picoseconds with the same frequency as the number of defective waters in the system. Moreover, the oscillations seem to be correlated in time: the larger the number of the defective water molecules, the more large-amplitude swings simultaneously happening in the system. Although it has been well appreciated that a sizable fraction of the hydrogen-bond network involve the presence of coordination defects[21,22], their role in reorientational dynamics has remained elusive. In fact, in panels b and c of Fig. 3, we show that when the local topology in the H-bond network is more defective, there are less small-amplitude angular changes in the system and more of the large-amplitude ones. Supplementary Fig. 2 illustrates similar analysis for swings with $\Delta\theta > 40°$.

Our findings suggest a deep connection between the fluctuations in the underlying local topology of the H-bond network and the nature of angular swings occurring simultaneously in the system. These results reinforce a picture of water reorientations being an outcome of highly coordinated dynamics of water molecules, rooted in the collective fluctuations of the network's topology consistent with previous

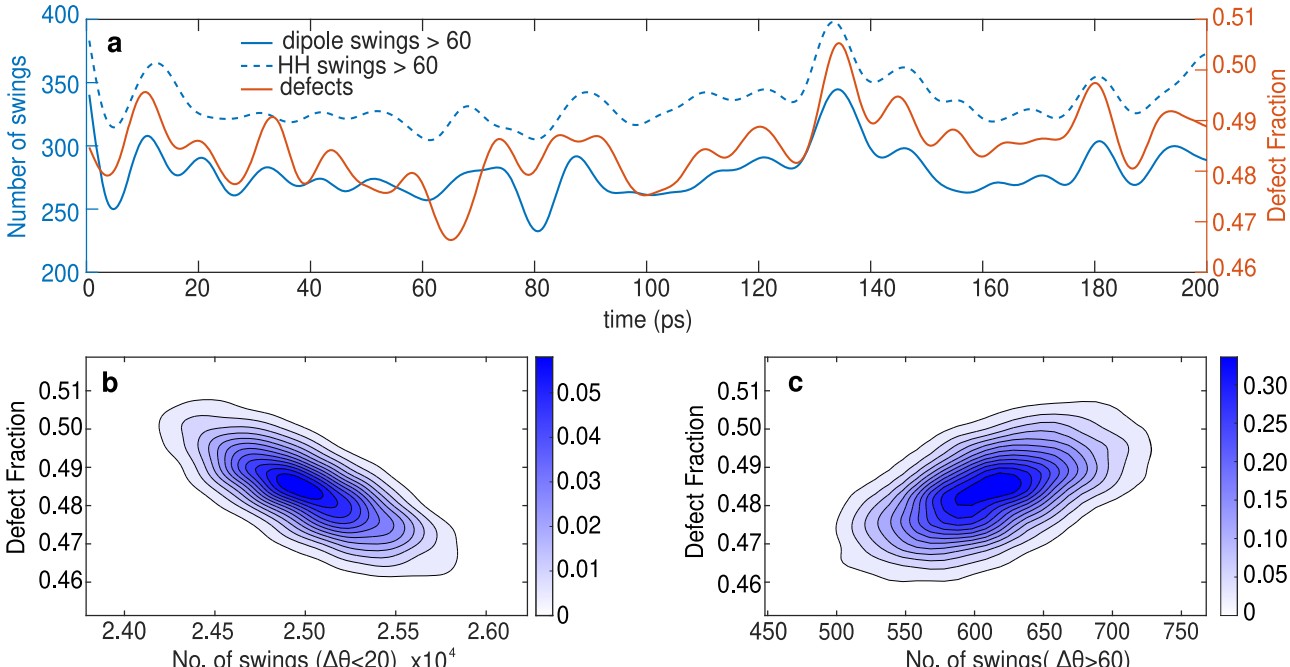

**Fig. 3 | Correlation between the number of simultaneous large angular swings and the fluctuations in the local topology of the water H-bond network. a** Time series of the number of molecules in the H-bond network performing large angular swings (amplitude larger than 60 degrees) at each moment of time as detected from the observation of the dipole vector (blue full line) and HH vector (blue dashed line). At each moment of time, we count the number of swings happening in the system within a time window of 1ps around it. We superimpose these time series with the time series of the fraction of molecules in the H-bond network that are defective, i.e., with non-tetrahedral local topology (red). We observe fluctuations of the order of tens of picoseconds in all three curves that often appear to be correlated in time. **b** Density plot of the fraction of defects in the H-bond network with respect to the number of molecules in the network performing small angular swings ($\Delta\Theta < 20$) within 1 ps. Anti-correlation between these two quantities means that when there are more molecules with defective local topology, the less small-amplitude angular swings occur in the H-bond network. We find the correlation coefficient to be $-0.7390 \pm 0.0089$, with $p < 0.01$. **c** Density plot of the fraction of defects in the H-bond network with respect to the number of large-amplitude angular swings ($\Delta\Theta > 60$) within 1 ps. Correlation between these two quantities indicates that the more the local topology in the H-bond network is defective, the larger is the number of molecules that perform large-amplitude angular swings. The correlation coefficient found is $0.5604 \pm 0.0151$, with $p < 0.01$.

proposals by Ohmine and co-workers[23]. Our findings bare some similarities with recent work by Liu and co-workers[24,25] where they show that an angular jump of a given water molecule could enhance the subsequent jump motions of the same water molecule and surrounding water molecules up to the 2nd coordination shell.

Perhaps not surprisingly, when looking at how correlated in time the angular trajectories of two molecules performing large jumps *concurrently* are, we find high correlation on the timescale of 10s of femtoseconds encompassing these angular jumps. However, as we look at the correlations between trajectories of the same molecules over longer times, they become less and less correlated akin to what is expected for any two random molecules in the system (see Supplementary Note 1 and Supplementary Fig. 3). While this confirms that there is indeed a high number of large jumps performed simultaneously by molecules, it still does not explain the collective nature and the mechanism underlying these events.

### Spatial correlations of jumping waters

Clues into the origins of the collective reorientational dynamics can be seen in Fig. 1a which suggests that many of the highlighted molecules performing large jumps close by in time, also seem to be close by in space, as if forming clusters of concurrently jumping molecules. This suggests that underlying these neighboring large jumps there is a coordinated reorientational dynamics facilitated by the reorganization of the local H-bond network. The specific details of the spatial distribution of the water molecules that simultaneously perform large angular jumps in the network will be tuned by both the timescales and magnitude of the reorientational dynamics. An important outcome of our automated protocol illustrated earlier, is that it gives direct

insights into this information through the statistics of $\Delta t$ as a function of $\Delta\theta$.

Figure 4a shows the probability distributions for the duration of swings with the amplitude $\Delta\Theta$ larger than a certain angular threshold for the DP vector of the water molecules. For small angular thresholds, when the selected swings are predominantly of small magnitude, the swing duration peaks at roughly 30 fs. This corresponds to a fast hindered rotational mode. As we increase the angular magnitude threshold, we find that the probability distributions change, both for the DP and the HH vectors (see Supplementary Fig. 4a). For intermediate values of the angular amplitude, a second peak in the probability distribution appears and the distribution becomes bimodal. A new characteristic time of around 100 fs emerges when looking only at angular fluctuations with the magnitude larger than 60°, where the second peak in the bimodal distribution becomes prominent. This in turn corresponds to the characteristic time of large angular swings. Note that for the duration of the jump we consider just the angular reorientation of the water molecule, and not the time needed for the H-bond breakage and formation, as often considered in the literature[9,10]. The characteristic time of 100 fs that emerges from our automated protocol instead, corresponds more closely to the time it takes to transition from one stable hydrogen-bonded state to another.

With this information on the characteristic time, we first determined the total number of jumps that occur within a time interval of 200 fs shifting the time window along the entire trajectory. Figure 4b shows the fraction of jumping water molecules in the ensemble obtained for two different $\Delta\theta$ thresholds. Interestingly, out of ~1000 waters in our simulation box, our detection protocol shows that approximately 100 water molecules undergo jumps with $\Delta\theta > 60°$. As

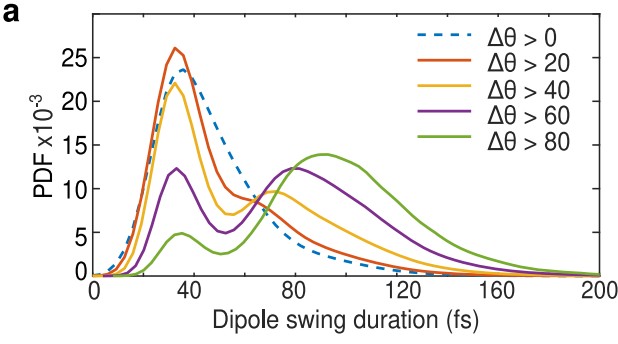

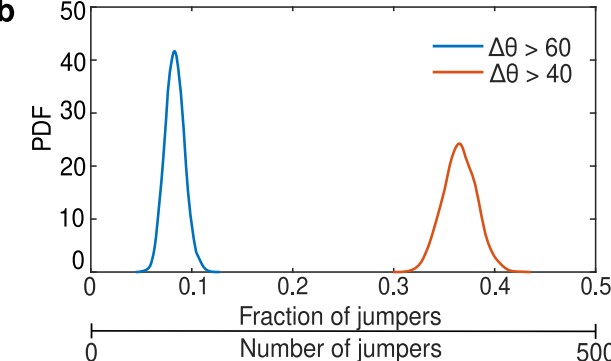

**Fig. 4 | Duration and collectiveness of angular swings. a** Probability distributions of duration $\Delta t$ of swings with the angular magnitude $\Delta\Theta$ greater than a certain threshold, detected from the DP vector time series. **b** Probability distribution function of the number of molecules in the system that perform large angular swings within a time window of 200 fs, which are considered to be occurring concurrently. Average number of large swings with the magnitude $\Delta\Theta > 60°$ within the selected time window is around 100, which is around 10% of the total number of molecules. Instead, the number of large reorientations with $\Delta\Theta > 40°$ is around 350, around 35% of the total number of water molecules.

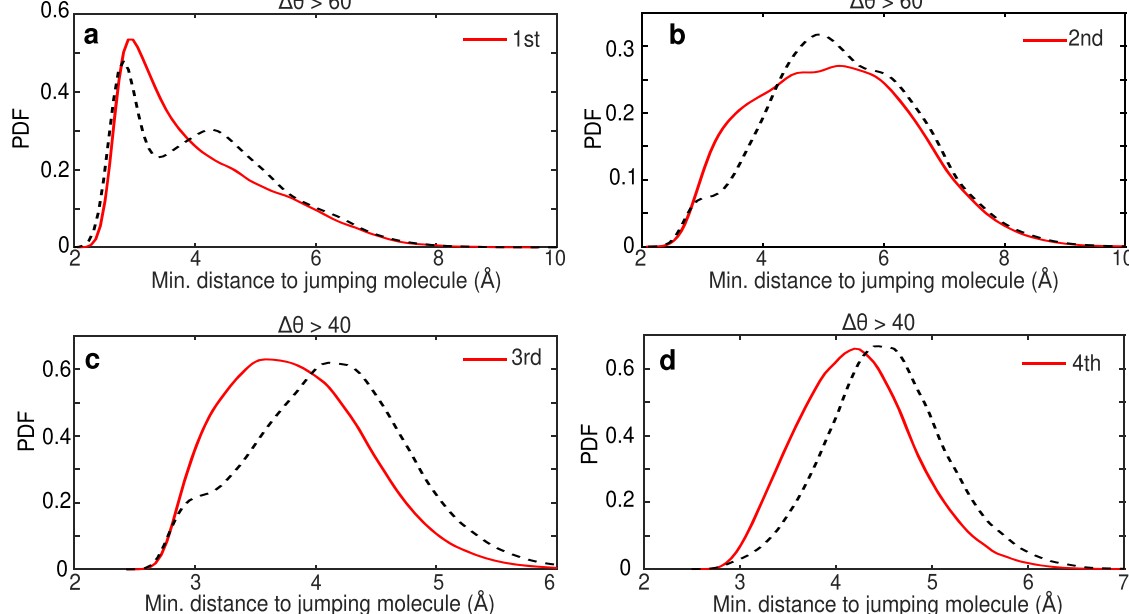

**Fig. 5 | Large angular swings occur close by in space. a–d** For the water molecules performing large swings within the selected time interval of 200 fs, full red lines show the probability distribution function of the distances to the first 4 nearest jumping molecules with respect to another jumping molecule. Dashed lines show the same probability distribution functions of the distances to the first 4 nearest waters when the water molecules were randomly selected from the whole ensemble. We select the same number of random waters in the system as there are large jumping water molecules within the same time window. In order to highlight the extent of spatial correlations between jumping molecules, we consider large swings with amplitudes $\Delta\Theta > 60°$ in panels (**a, b**) and with $\Delta\Theta > 40°$ in **c, d**. PDFs for both thresholds and all 4 nearest neighbors are given in Supplementary Figs. 8 and 9. The distributions are constructed using a kernel histogram. The kernel width used for constructing the smoothened distributions ranges between 0.1 and 0.2. We also performed statistical tests (Kolmogorov-Smirnov[62]) that reinforce the statistical significance of the differences in the distributions we show. The Kolmogorov-Smirnov tests reject the null-hypothesis that both distributions (jumping and random) are sampled from the same distribution with a 5% significance level with p-value significantly smaller than 0.01.

expected, relaxing the threshold of the jump magnitude down to 40° enhances the population undergoing angular transitions to ~350 water molecules. If such a large number of water molecules undergo angular fluctuations, do they occur independently from each other or are there some underlying connections in the hydrogen bond network involved in these reorientational motions?

To address this question, we analyzed the spatial position of the strongly jumping molecules with respect to each other by calculating the distances to the 1st, 2nd, 3rd, and 4th nearest molecule that jumps within the selected time interval. We obtain the probability distribution function (PDF) for these distances by repeating the analysis for the time interval of 200 fs, shifting the time window along the whole duration of the simulation. The obtained PDFs for the position of the nearest jumping molecules with respect to another jumping molecule are shown in Fig. 5, when looking at the angular jumps of magnitude larger than 60° and 40°, respectively. To understand whether the jumps are distributed homogeneously in space or not, we compare these PDFs to the case where, instead of the concurrently jumping molecules, we look at the same number of randomly selected molecules in the network. The PDFs for the distances of the first 4 nearest random molecules to another random molecule are shown by dashed lines in Fig. 5.

For angular jumps with $\Delta\Theta > 60°$, the difference in the PDFs is most evident for the 1st and 2nd nearest neighbor (Fig. 5a, b).

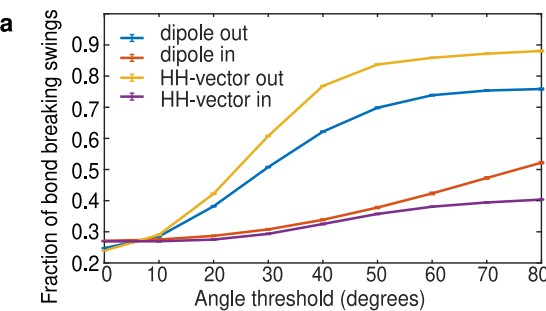
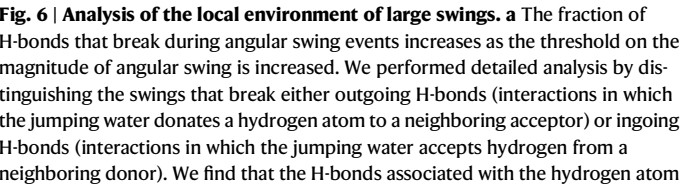
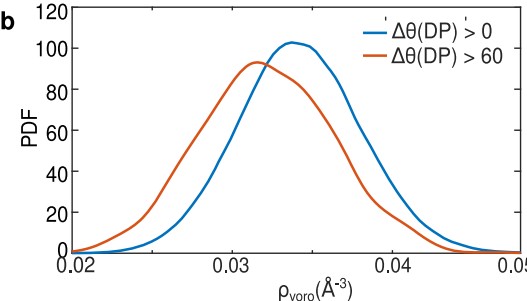

**Fig. 6 | Analysis of the local environment of large swings. a** The fraction of H-bonds that break during angular swing events increases as the threshold on the magnitude of angular swing is increased. We performed detailed analysis by distinguishing the swings that break either outgoing H-bonds (interactions in which the jumping water donates a hydrogen atom to a neighboring acceptor) or ingoing H-bonds (interactions in which the jumping water accepts hydrogen from a neighboring donor). We find that the H-bonds associated with the hydrogen atom of the swinging molecule (outgoing blue and yellow) is affected the most by the large angular swing of the water. As before, we distinguish the swings detected from the DP and from the HH vector time series (see Supplementary Fig. 4b for the total fraction of bond breaking swings). **b** Probability distributions of the Voronoi density, $\rho_{voro}$, for the water molecules undergoing angular swings constrained by a magnitude threshold. The probability distributions shift towards lower local densities as we restrict ourselves to swings with the larger angular magnitude.

The nearest jumping molecule is typically situated in the first water shell of another molecule that performs large jump, while the 2nd nearest neighbor has a wide distribution around the second water shell distance. Instead, if the concurrently jumping molecules would be distributed homogeneously throughout the system, as is the case when choosing random molecules, the two nearest neighbors would have been positioned at larger distances with a rather different shape of the PDFs (dashed black lines in Fig. 5a, b). This gives a strong indication that the simultaneous large jumps we observe are not happening independently throughout the system, but are instead interconnected and correlated.

The collective re-orientational motions that we elucidate in our work, involves a spectrum of transitions of both the timescales and angular magnitude, as seen in Fig. 4a. Thus, when angular jumps occur in a correlated fashion they will naturally involve a combination of jumps of different magnitude. Although jumping and random spatial distributions for the 3rd and 4th nearest neighbors exhibit dissimilarities also for the jumps with larger angle threshold of 60°, the differences are even more pronounced for the lower angle threshold of 40°. Indeed, when we include the large jumps with a somewhat smaller amplitude, $\Delta\Theta > 40°$, we see from the PDFs that the 3rd and 4th jumping nearest neighbors are situated much closer in space to the jumping molecules than if they would have been randomly distributed (see Fig. 5c, d). This means that the large jumps generate a myriad of relatively large reorientations that occur in spatially organized clusters due to the local H-bond network rearrangements.

**Hydrogen bonding network: topology and density fluctuations**
With all this dynamical activity in the hydrogen bond network involving highly cooperative processes, we can now return back to how this complex behavior is reflected in changes in the local water structure. Fluctuations of water can involve changes in local topology[26], coordination structure[27], and density[28]. Laage and Hynes for example, showed that rotating water molecules break hydrogen bonds with overcoordinated neighbors and subsequently form interactions with undercoordinated water molecules[9,10]. At the same time, some studies have also shown that modulations in the local density can also affect the water dynamics[29]. Since our automated protocol detects angular swings without initial imposition of hydrogen bonding interactions, it is interesting to examine the correlation between the collective reorientational dynamics and changes in the number of hydrogen bonds. We next investigate how many of the detected angular swings are actually bond breaking events, where we count all the events which have a change in the H-bonded neighbors before and after a jump.

Figure 6a illustrates that as one increases our threshold on the magnitude of the angular swing, the probability that this incurs a hydrogen-bond breaking event is enhanced. This feature is observed for both the DP and HH vectors. Interestingly, there also appears to be an asymmetry in the behavior of outgoing (donating hydrogen bonds) versus incoming (hydrogen bonds being accepted by the jumping water). Large swings associated with the HH vector break donating H-bonds roughly 90% of the time, while accepting H-bonds are broken 40% of the time. In the case of the DP vector, this asymmetry persists, although to a lesser degree (70% for out and 40% in). The difference in behavior of large swings between donating versus accepting hydrogen bonds is essentially rooted in the position of the proton along the respective hydrogen bond. In the case of donating hydrogen bonds, angular swings require a large change in the O-H bond. On the other hand, for accepting hydrogen bonds, it is possible to make reorientational motions without necessarily disturbing the orientation of the O-H bond from another water molecule that is further away.

Indeed, this is also consistent with the original large jump mechanism proposed by Laage and Hynes, which involves an exchange of the H-bonds acceptors[9,10]. Here, however, by using the automatized detection of the angular swings, without a priori selecting the events based on the H-bond breaking, we are able to uncover a multitude of the swinging events occurring in the system that are interrelated by the underlying H-bond network rearrangement. Importantly, the asymmetric nature of the H-bond breaking events will be collectively manifested as an anisotropic diffusion of the topological defects.

Another interesting aspect of Fig. 6a is that it shows that a sizeable fraction of large angular swings involve the breaking of hydrogen bonds that are donated to the water molecule under study. For the dipole vector, approximately 50% of the large jumps involve hydrogen bonds being broken on the donating side. This feature is consistent with the collective nature of the swings elucidated earlier. The breaking of hydrogen bonds being donated to jumping waters implies that there are at least several pairs of water molecules that undergo simultaneous reorientational motions.

Besides the changes in local coordination topology dictated by direction of the hydrogen bonds, density fluctuations are central to understanding both the thermodynamic and dynamical properties of water[30,31]. We speculated that regions of the hydrogen bond network which are lower in density might be more susceptible to reorientational motions. For all the identified jump time intervals, we select the mid-points and studied the statistics of the Voronoi density[32,33] ($\rho_{voro}$) defined as the inverse of the Voronoi volume taken as a sum of the oxygen and two hydrogens of the jumping waters. Indeed, Fig. 6b shows that as we increase the threshold of the magnitude of the

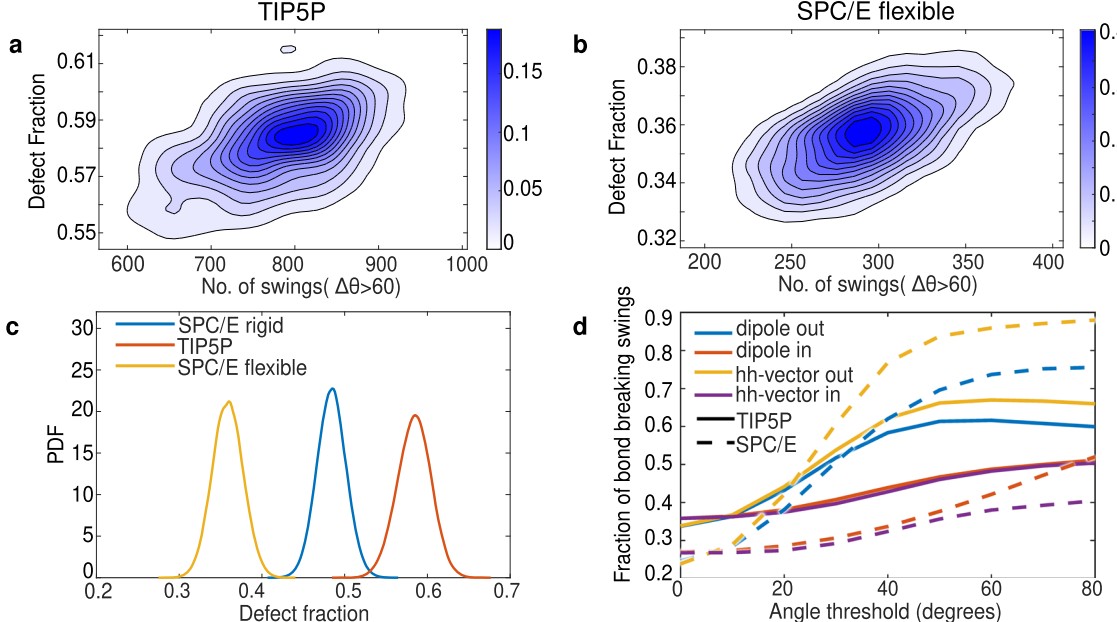

**Fig. 7 | Collective burst mechanism for different point-charge water models.** **a**, **b** Density plots of fraction of defects in the H-bond network versus the number of large jumps within 1ps for TIP5P water model and SPC/E flexible water model, respectively. The correlation coefficients found are **a** 0.4596 and **b** 0.50. **c** Probability distribution function of the fraction of defects for different water models. **d** Fraction of H-bonds that break on the donating vs accepting side, as a function of the magnitude of angular swing for SPC/E and TIP5P water models.

angular jumps, we observe that the peak in the distribution of the Voronoi density shifts towards smaller values. This shows that large angular swings are enhanced in low density environments. The apparent relationship between the magnitude of angular jumps and the local-density fluctuations suggests a possible coupling between the fluctuations of the network involving translational and rotational degrees of freedom. It would be interesting in future studies to examine this effect in supercooled water as recently highlighted by Martelli[34].

As seen from the previous analysis on the changes in the local environment involving the local topology of the hydrogen bond network, large swings must naturally result in the disruption of the hydrogen bond network of at least one of the nearest neighbors. Furthermore, large angular swings also tend to occur in low-density and more disordered environments. Since density fluctuations involve collective reorganizational processes within the hydrogen bond network, large swings could presumably either lead to consequent angular jumps of the nearby molecules, as suggested in the literature[24,35,36] or alternatively, be part of several large angular swings that occur simultaneously.

**Burst mechanism and water model sensitivity**

One might wonder how much the results we find are affected by the size of the system we consider. In order to discard spurious finite size effects on the collective reorientation mechanism proposed here, we conducted a series of additional simulations for larger system sizes and analyzed their large swings and defect statistics (for details see Methods section, Supplementary Note 2, and Supplementary Figs. 6-9). We find that, while the relative magnitude of the large swings and defects fluctuations decreases with the system size, effectively disappearing in the thermodynamic limit, the picture of the reorientation mechanism still persists even for larger systems. Indeed, what is critical for the collective reorientation dynamics is that, also for the large systems, the number of defects and large swings occurring locally in the system are still correlated, and the spatial correlations between the jumping and the random molecules are not sensitive to finite size effects.

In order to examine the sensitivity of our collective orientational mechanism to the choice of the water model, we compared the rigid SPC/E to both the flexible version of SPC/E[37] and the TIP5P water model[38]. The flexible version of SPC/E has been shown to bring many of the structural and dynamical properties of liquid water into closer agreement with the experiments. Remsing and co-workers recently demonstrated the electrostatic potential in the center of a cavity is rather sensitive to the choice of water model. Specifically, they showed that three-point-charge models induce a bigger asymmetry in this potential on donating vs accepting side of hydrogen bonds compared to the five-site model TIP5P that appears to be in close agreement with the DFT-based simulations[39]. Comparing SPC/E and TIP5P water models in terms of the mechanisms we introduce in this work, provides a way to probe limiting cases of how the lone-pair electrons are treated within point charge water models[40]. We thus focused our efforts on comparing our results from SPC/E to TIP5P circumventing the challenges of using DFT-based ab initio molecular dynamics. Although we were limited to small system sizes, we also find using the many-body polarizable water model, MB-pol[41,42], that the jumps also come in waves (see Supplementary Fig. 20).

Our analysis shows that for both the TIP5P and flexible SPC/E water models, one observes a correlation between the number of jumps and defects as seen in panels a and b of Fig. 7 (see also Supplementary Note 3). Similarly, both water models also exhibit the same trends for the differences between spatial correlations observed for the jumping versus random water molecules (Supplementary Figs. 11-12, 18-19). In the case of TIP5P, jumping water molecules tend to be found clustered slightly closer together in space compared to SPC/E, while for the flexible version of SPC/E, the effect is reversed, with jumping waters clustered further away compared to what is found in the rigid model.

To understand better how the different water models affect the collective reorientational mechanism, we also compared the distribution of the number of defects for the three water models which is shown in Fig. 7c. Interestingly, the fraction of defects is TIP5P is enhanced relative to SPC/E while in the flexible version of the SPC/E model, the concentration of defects is reduced. For TIP5P, we observe

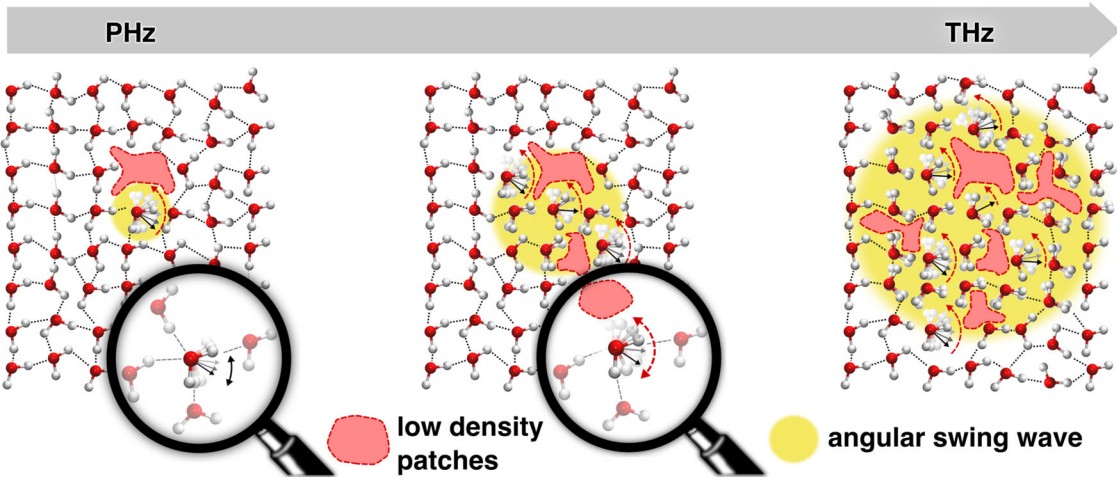

**Fig. 8 | The scheme depicting the essential ingredients associated with the collective angular jump mechanism in liquid water.** Moving from left to right, large angular jumps near low-density regions of the water network, create a wave of jumps that propagates, as depicted by the growing yellow circle, to many different water molecules. Waters that are then close to low-density patches are also susceptible to large jumps. All in all, a highly collective process from the peta-to-teraHertz (PHz to THz) timescale occurs involving changes in topology and density triggered by the angular jumps.

an enhancement in the concentration of the 1in-2out, 2in-1out, and also the 1in-1out defect water molecules (Supplementary Fig. 14). This is not too surprising since the presence of the additional sites in the TIP5P model, analogous to the lone-pair Wannier centers in DFT, enthalpically stabilizes hydrogen bonds that would otherwise be less stable in a 3-site model. This enhancement of the concentration of the defects serves as a seed for observing a larger number of swings in the TIP5P model.

The enhancement of the collective reorientations in the TIP5P water model can also be observed in the fraction of hydrogen bonds that break during swing events. Figure 7d compares the asymmetry in the number of hydrogen bonds that break on the donating vs accepting side, as a function of the magnitude of the angular swing for the SPC/E and TIP5P water model. In the TIP5P water model, there is a larger number of swings where water molecules on the donating side also involve hydrogen bond breaking events. In other words, there are more pairs of water molecules that undergo jumps.

For the case of flexible SPC/E, the fraction of defects decreases relative to the rigid model, see Fig. 7c (also see Supplementary Fig. 17). In the flexible water model, the O-H bonds being more delocalized enhances the strength of the hydrogen bonds and therefore reduces the concentration of defects from approximately 48% to 37%. Due to this difference, jumping (and therefore defective) water molecules are spatially correlated farther away from each other than in the rigid model.

## Discussion

In the present work, we elucidated the mechanism and origins of the collective reorientation in bulk water by employing molecular dynamics simulations and an automated detection of abrupt angular fluctuations. The automated protocol that detects large changes in the orientational motion of water molecules does not depend on any a priori criteria for hydrogen-bond interactions and allows identification of diverse non-local orientational swings.

The picture that emerges from our communication is illustrated in the schematic shown in Fig. 8. Essentially large angular jumps can occur at certain hot-spot regions, such as near low-density regions conferring greater rotational mobility to the water molecules. Due to the correlations in the hydrogen-bond network involving fluctuations in both topology and density, large jumps do not happen in isolation and subsequently, over the peta-to-TeraHertz timescale, we observe waves of both large and small jumps that happen concurrently.

Although these dynamics have been observed in several previous spectroscopy-based experiments[15,43,44], the collective origin of these modes has remained poorly understood.

One of the reasons why the interpretation of microscopic water dynamics is challenging even in the presence of large amount of simulation results is that the timescales of slow single-water reorientation processes are similar to those of the collective dynamics of many waters[45]. While all the previous studies considered the concerted rotation of three water molecules while examining the reorientation dynamics, we have shown that the collective reorientation and H-bond network rearrangements involve several larger groups of near-by molecules that are spread throughout the system. The spatial extent of the correlations and their coupling to thermal density fluctuations that occur on similar timescales[30] would be interesting to study in the future. In addition, the dielectric relaxation of water has been shown in several studies to involve the collective relaxation of water dipoles[46,47]. The non-local reorientation mechanism we elucidate here is consistent with this picture.

Our findings on the collective nature of angular jumps in bulk water have important implications and connections with previous studies inspecting the properties of water across the phase diagram. In recent years, Martelli and co-workers have quantified the complexity and topology of the hydrogen-bond network in terms of both ring-statistics[48] and topological defects akin to our analysis, both at room temperature and under supercooled conditions[34]. In addition, Di Stasio and co-workers have studied using ab initio molecular dynamics the role of electronic structure in altering the relative population of topological defects in bulk water[49]. A natural extension of our work, would be to explore how the collective reorientational dynamics change under supercooled conditions although this would require significantly longer simulations to converge. This may be an ideal playground for the use of deep neural networks[50] as well as many-body polarizable potentials[41,42] where the role of electronic effects and polarization on the reorientational mechanism can be investigated in the future.

Finally, our numerical results and molecular insights should motivate the creation of theoretical models to describe the cooperative dynamics in hydrogen-bonded liquids[51] which may play an important role in tuning chemical reactions[52]. The large angular swings we observe are akin to the tunneling dynamic pathways of water clusters at low temperature[53] which opens up interesting connections between the mechanisms of reorientational dynamics at low

temperature and in the condensed phase. Although we have focused on bulk water at room temperature showing the highly cooperative character of the reorientational dynamics of the water molecules, we believe that our results open the doors to exploring how this effect changes upon supercooling and near biological systems where one might expect an enhancement of the phenomenon. Very recent results by Laage and coworkers suggest that the HB-breaking angular events are related to water translational diffusion, pinpointing the connection between water's collective reorientation and its transport properties[54].

## Methods

### Molecular dynamics

We performed a molecular dynamics simulation of 1019 water molecules using the GROMACS 5.0 package[55] with the SPC/E rigid water model[56]. This water model was used previously by Laage and Hynes[10]. Energy minimization was first carried out to relax the system, followed by an NPT and subsequently NVT equilibration at 300 K and 1 atmosphere for 10 ns each. A timestep of 1 fs is used for all the simulations. The NVT simulations were performed using the canonical velocity-rescaling thermostat[57] with a time constant of 2 ps. The choice of the thermostat also ensures that dynamical properties of water are not disturbed. The NPT runs were conducted using the Parrinello-Rahman[58] barostat with a pressure coupling time constant of 2 ps. The production run at 300K was carried out for 2 ns in the NVT ensemble where the trajectory was outputted every 4 fs in order to resolve the dynamics associated with the fast reorientations. The dimensions of the cubic box used for the NVT simulations were 31.1970 Å.

We have also performed simulations and repeated several of our analysis for larger system sizes, namely for one consisting of 3400 (box size 46.720 Å) and another of 8152 water molecules (box size 62.5572 Å) in cubic periodic boxes. In addition, to investigate the generality of the collective reorientational mechanism, we conducted small-box simulations for several point-charge water models, namely TIP5P[39] and flexible SPC/E[37] water models. The results for these are shown in the Results section and in Supplementary Note 3. In Supplementary Fig. 20, we also compare some of our results with the MB-pol potential which is among the most accurate in-silico potentials reproducing many of the properties of water across the phase diagram[41,42].

### Automatized detection of angular swings

Water reorientation dynamics include various processes happening at different time scales, from very fast vibrational motions causing limited reorientation, to slower reorientation through sudden large-amplitude angular jumps. It remains still an open question how and to which extent each of these processes is involved in the underlying collective hydrogen bond rearrangements. To elucidate the mechanism behind the collective reorganization of water observed in Fig. 1, we developed an automatized protocol for detecting all the various angular changes in water reorientation, which we term angular swings. Our protocol identifies these angular swings without any prior knowledge nor by using geometric or energetic criteria for the hydrogen bonding interactions. Here, we describe all the steps of the protocol which is summarized in Fig. 2.

In order to track down angular changes in water orientation, we rely on two body-fixed vectors, the dipole moment (DP) and the HH vector (see Fig. 2a). From the MD simulation, we first extracted the DP and HH vector time series for each water molecule, herein referred to as $\mathbf{v}(t)$, with $t$ being time. A new times series, $\mathbf{v}_F(t)$, was constructed by filtering $\mathbf{v}(t)$ using a second-order low-pass digital butterworth filter[59] implemented in MATLAB[60] with the cutoff frequency of 10 THz. To further smooth the time series, a mean filter of 10 THz was then subsequently applied. Details of the filtering procedures are specified in the following Methods section. An example of the original unfiltered

and the filtered time series of one of the DP vector components are shown in Fig. 2a, where we can observe very clear sudden changes, such is the one at around time 1500 fs, that should be detected by the automatized protocol.

Using the filtered time series we subsequently compute the derivative of $\mathbf{v}_F(t)$ using a finite difference method, and calculate the cross product of this vector with $\mathbf{v}_F(t)$ in order to obtain a new vector

$$\mathbf{n}(t) = \mathbf{v}_F(t) \times \frac{d\mathbf{v}_F(t)}{dt}, \tag{1}$$

which corresponds to the vector perpendicular to the plane of rotation of the body-fixed vector $\mathbf{v}_F(t)$. Our protocol defines an angular swing to be a process that does not change the plane of rotation of the body-fixed vector $\mathbf{v}_F(t)$. This implies that, over the time of one angular swing, the direction of $\mathbf{n}(t)$ does not change, as shown in Fig. 2b. The start and the end points of swing events are then identified as large instantaneous changes in the direction of $\mathbf{n}(t)$. More precisely, we look at the following quantity

$$q(t) = 1 - \frac{\mathbf{n}(t) \cdot \mathbf{n}(t+dt)}{|\mathbf{n}(t)||\mathbf{n}(t+dt)|}, \tag{2}$$

which is equal to 0 during the swing. At the start and at the end of the swing, this quantity is found to be non-zero, indicating the change in the plane of rotation of the body-fixed vector $\mathbf{v}_F(t)$. Consequently, the start and end times of angular swings can be identified as maxima in $q(t)$, which are fully consistent with extrema in the filtered time series, $\mathbf{v}_F(t)$, correctly recognizing sudden angular changes as shown in Fig. 2a.

Finally, having identified the start and end points of the swings, the duration of the swings are taken to be the times between two peaks of $q(t)$, and the magnitude is found by computing the angle between the unfiltered DP or HH vector at the start and at the end point of the swing. The final output of the protocol is the start time (t), duration ($\Delta t$), and magnitude ($\Delta \Theta$), for each angular swing detected. We perform the procedure both for DP and HH vectors, as some angular fluctuations of water molecules can be better captured through one or the other vector.

### Filtering of the time series

On the time traces of the molecular dipole moments and HH-vectors, we first applied a 2nd order low-pass digital butterworth filter (MATLAB implemented, designed to have a frequency response as flat as possible in the passband). The trajectories are sampled every 4 fs, while the cutoff frequency (the frequency at which the magnitude response of the filter is $1/\sqrt{2}$) which corresponds to 10 THz (1/100 in inverse fs). In order to further smoothen the trajectories before applying the automatized detection of the angular swings, we applied a mean filter using a MATLAB function *smooth* with the span of 25 data points (corresponding to the time window of 100 fs). Furthermore, during post-processing of the time series of defect fractions and number of swings, we have additionally applied a low-pass butterworth filter with the cutoff frequency of 0.1 THz (1/10000 in inverse fs), which allows us to clearly capture the fluctuations on the 10 ps scale. The same smoothing procedures were applied for all the water models and system sizes.

### Hydrogen-bond definition

To determine the hydrogen bonds, a geometrical criterion is used developed originally by Luzar and Chandler[61]. A pair of water molecules are considered to be hydrogen bonded when the distance between the donating and accepting oxygen atoms ($O_D$ and $O_A$) is less than or equal to 3.5 Å and the angle formed by the bond vector

between the donating hydrogen and oxygen ($H_D$ and $O_D$) and that between $O_D$ and $O_A$ is less than 30˚.

## Data availability
The data that support the findings of this study are available in the Zenodo repository: https://doi.org/10.5281/zenodo.7646486.

## Code availability
The codes that were used for this study are available in the Zenodo repository: https://doi.org/10.5281/zenodo.7646486.

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

## Acknowledgements

A.H. acknowledges the funding received by the European Research Council (ERC) under the European Union's Horizon 2020 research and innovation programme (grant number 101043272 - HyBOP). Views and opinions expressed are however those of the author(s) only and do not necessarily reflect those of the European Union or the European Research Council Executive Agency. Neither the European Union nor the granting authority can be held responsible for them.

## Author contributions

U.N.M., A.R., A.H., and A.J. designed research; A.O.D., U.N.M, A.R., A.H., and A.J. performed research; A.O.D. contributed new method tool; U.N.M., A.R., A.H., and A.J. wrote the paper.

## Competing interests

The authors declare no competing interests.
