## [Peer review file · Nature Communications]

REVIEWER COMMENTS

Reviewer #1 (Remarks to the Author):

The manuscript by Offei-Danso et al. describes a detailed analysis of a molecular dynamics simulation of liquid water, which unveils new insights into the intricate details of water orientational dynamics. Earlier work by Laage & Hynes identified the existence of large angular jumps that primarily determine the decay of orientational correlation functions in water instead of sequences of small rotational diffusional events.

The authors in the present study propose an interesting approach to identify and analyze orientational jumps or swings, which is innovative and allows for a clean separation and characterization of individual events involving orientational motion of water molecules. This allows them to show that large amplitude orientational swings are not performed by isolated molecules, but occur collectively in water molecules within close proximity and are correlated with local defects in the water hydrogen bond network and low local densities.

One concern that I have is that some of the effects, e.g. the oscillations of the total number of large amplitude swings within the entire system as a function of time, are indicative of finite size effects for the small system that is being simulated and analyzed here. Clearly, no such oscillations are expected for macroscopic samples. I don't expect this to affect the main conclusions, but finite size effects and their possible impacts on correlated dynamics should be discussed in more detail.

The main conclusions are of interest for the general topic of dynamical processes in water and aqueous solutions, which are probed directly or indirectly by multiple spectroscopies. Thus, the results are relevant for a broad audience and may be suitable for Nature Communications.

In addition to the lack of a sufficient discussion of finite size effects, I have several comments that are listed below and should be addressed prior to publication:

1. The abstract advertises this work as an example of "unsupervised learning", which seems to attempt to use language from the machine learning community as a buzz word. However, what is presented is a deterministic analysis of atomistic molecular dynamics simulations and does not involve any "learning" algorithm. Hence, the term "unsupervised learning" should be removed because it is misleading potential readers.

2. The use of low-pass and smoothing filters (what kind of smoothing filter? Gaussian? Blackman-Harris?) in the frequency domain is explained for the analysis of time traces of the molecular dipole moments and HH-vectors. However, filters are evidently also used to post-process time traces of defect fractions in Figs. 3, S1, S2, S7. Comparisons between Fig. 3 and S1 indicate that different filters are used in different plots of the same property, which is highly confusing especially since the filters aren't even mentioned in this context.

3. To define defects in the hydrogen bond network, a geometrical or energetic criterion for the definition of a hydrogen bond has been used, which is not described in the manuscript.

4. I found Figure 5 highly confusing. It seems very strange that distance distributions are shown for the closest two water molecules for large orientational swings, while the distance distributions of the 3rd and 4th closest water molecules are shown for smaller swings. This is very difficult to follow and the reason for this choice is unclear. Fig. S4 shows the complete data set to some degree but the cumulative probability plots are not very useful in this case, since by definition all probabilities add up to 1 and the details of the distribution functions are obfuscated. I recommend to replace Fig. 5 with a version of Fig. S4 that shows the distribution functions instead of cumulative probabilities.

5. What further puzzles me is the smoothness of the distribution functions in Figure 5. Given that the simulation only contains ~1000 water molecules and the statistics are obtained for just 2 nanoseconds, distance distributions between only water molecules that perform large orientational swings within the same 200fs time window should have some noticeable statistical limitations. What bin size was used to compute the distribution functions? Was any smoothing operation applied to the shown curves? (such details were apparently omitted for other plots in the manuscript, as mentioned above, which is why I suspect them here)

Minor comments:

6. Abstract: ThZ should be THz

7. Page 7, paragraph 2, first word: "Armored" should be "Armed". In any case, a less colloquial formulation may be more suitable

Reviewer #2 (Remarks to the Author):

I have read the article entitled, "THE COLLECTIVE BURST MECHANISM OF ANGULAR JUMPS IN LIQUID WATER" with great interest. It is a careful study regarding the how the hydrogen bond network reorients using a ML model that relates large reorientations to collective motions. In some sense, it is not surprising that large reorientations require collective motions. Nevertheless quantifying these "bursts" is interesting and important. This study puts a number to this and this is somewhat interesting.

I like Figure 7.

I think the authors have done an amazing job at quantifying and showing correlations between large rotational swings and the underlying collective structure. They do talk about 2 different water models, but fail to really describe if there are quantitative or qualitative differences between the two models. Obviously, an article in the Nature class journals needs to teach the reader something new about water. I am not sure in its present form this article passes this standard. Don't get me wrong, this paper should be published and will likely be highly cited as the water community is large and diverse in its interests.

I would have liked to see how these jumps change as a description of the local structure--MB-pol vs SPC/E even DFT-based. The class of models such as SPC/E are classified as "frustrated charge" models. They do describe bulk aspects reasonably well (pretty good even). This paper is about details, and there are some important details to examine. To the extent MB-Pol is as charge frustrated as SPC/E is not clear. DFT based models have very "smooth" interactions and do not suffer from the rickety nature of the point charge models. Having an understanding about how this influence jump dynamics would be interesting and teach us something in lieu of an experiment. This is the classic problem of short-range vs long-range and how they are coupled. There is an opportunity to teach us about the nature of this coupling. Do all water models exhibit the same "burst' dynamics?

I would urge the authors to address these issues if they want to be in the Nature class journals. Otherwise this is a great technical study that should be accepted immediately in trade journal such as JCP or JPC.

Reviewer #3 (Remarks to the Author):

This is an interesting computational work aimed at understanding if angular jumps in water are correlated or not.

As the authors report, the goal of this work is definitely very high. The authors describe their approach quite nicely, and the article is well written, but I am not quite convinced about the soundness of the results.

In the following, I list the major points I would like the authors to address:

1. The authors performed simulations with 1019 water molecules described by the SPC-E model. It would be useful to show that the results are not affected by the system size and that they are general for various classical potentials. In the SI the authors report data for MB-POL, but not for 512 molecules. Therefore, I would not take them into account. While I understand the request of consistency in using the same model as in the original paper by Laage and Hynes, the SPC-E model is very poor. More realistic models need to be considered. In particular, it would be important to be sure that flexible models show similar jumps and analyse the existence of correlations.
2. Fig. 5 should be complemented with the RDF computed between 'jumping' molecules and compared to the random distribution. This is because the RDF for a random distribution is known, while the distributions reported in the current fig. 5 for random molecules are somewhat obscure.
3. Fig. 3A shows, even by eye, some appealing correlation. On the other hand, the same plot reported for MB-pol in the SI shows much less correlation, suggesting that polarisation and/or system size might play a crucial role.

Minor points:

1. The asymmetry between donors and acceptors in liquid water is quite known and well described in the literature. The authors should acknowledge previous works with appropriate citations (J. Chem. Phys. 141, 084502 (2014), J. Mol. Liq. 329, 115530 (2021), PNAS Nexus 1, 1-8 (2022)) and frame their results based on this previous knowledge.
2. Fig. 6: could be related to the decoupling of translational and rotational degrees of freedom occurring in the supercooled regime, as hypothesised in PNAS Nexus 1, 1-8 (2022). The authors should mention this.
3. The use of the word 'topology' in this manuscript is out of context, as the authors do not provide any analysis about the topology of the hydrogen bond network.
4. The authors refer to "Automated detection and unsupervised detection". Such wording infer the presence of a ML algorithm but it is not to be the case. The wording should be modified accordingly.

5. In the SI it is reported that large angular jumps occur in the order of 0.1 ps. On the other hand, density fluctuations occur on the order of 4ps (N.J. English and J.S. Tse, Phys. Rev. Lett. 106, 037801 2011, also cited by the authors), on a time scale comparable to that of hydrogen-bonded-related fluctuations (Santra et al., Mol. Phys. 113, 2829-2841, 2015).

In conclusion, I am not convinced that the 'jumps' are spatially correlated. The authors need to address my points before I can accept the manuscript for publication in nature communications.

Response to Reviewers

We thank the referees for their overall assessment of the quality and potential impact of our work. Following all the comments we received, we performed a substantial amount of further simulations and analysis. Below, we address all the issues they raised point by point, highlighting also all the changes that have been made to the manuscript. In particular, we have addressed the following questions:

- **Finite size effects:** we performed additional simulations for several larger system sizes. We find that the relative fluctuations in the number of defects and angular jumps decrease as a function of system size, as the Reviewer 1 suggested. Although these macroscopic fluctuations will effectively disappear in the thermodynamic limit, we find that the mechanism of collective large jumps still persists even for larger system size, as seen through the spatial correlations between the simultaneously jumping molecules, which occur at the length scales that are smaller than the system size.
- **Different water models:** we performed simulations of several models with different descriptions of local structures than the originally simulated rigid SPC/E water model, in particular, TIP5/P water model and flexible SPC/E water model. For all these water models, we find similar correlations between the number of large jumps and the number of defects in the system. Spatial correlations between jumping molecules also persist, ensuring the validity of the collective jump mechanism. Important differences between the water models and its implications on the mechanisms are also elucidated.
- We addressed all the other comments posed by the referees, as detailed in the responses.

The questions raised by the reviewers are reproduced below followed by our point-by-point responses and then the relevant changes that are added to the manuscript.

Reviewer 1 (Remarks to the Author):

The manuscript by Offei-Danso et al. describes a detailed analysis of a molecular dynamics simulation of liquid water, which unveils new insights into the intricate details of water orientational dynamics. Earlier work by Laage & Hynes identified the existence of large angular jumps that primarily determine the decay of orientational correlation functions in water instead of sequences of small rotational diffusional events. The authors in the present study propose an interesting approach to identify and analyze orientational jumps or swings, which is innovative and allows for a clean separation and characterization of individual events involving orientational motion of water molecules. This allows them to show that large amplitude orientational swings are not performed by isolated molecules, but occur collectively in water molecules within close proximity and are correlated with local defects in the water hydrogen bond network and low local densities.

One concern that I have is that some of the effects, e.g. the oscillations of the total number of large amplitude swings within the entire system as a function of time, are indicative of finite size effects for the small system that is being simulated and analyzed here. Clearly, no such oscillations are expected for macroscopic samples. I don't expect this to affect the main conclusions, but finite size effects and their possible impacts on correlated dynamics should be discussed in more detail.

The main conclusions are of interest for the general topic of dynamical processes in water and aqueous solutions, which are probed directly or indirectly by multiple spectroscopies. Thus, the results are relevant for a broad audience and may be suitable for Nature Communications.

Response:

We thank the reviewer for these comments and, in particular, the question regarding the role of finite-size effects on our mechanism of the collective and correlated orientational motions in liquid water. In order to assess the sensitivity of our results, in addition to our current simulations consisting of 1019 water molecules (box size 31.1970 Angstroms), we repeated our analysis for two other system sizes: one cubic periodic box consisting of 3400 (box size 46.720 Angstroms), and another of 8152 water molecules (box size 62.5572 Angstroms). Specifically, we compared how the relative fluctuations in the number of defects, which was found to be correlated with the number of jumps, changed as a function of system size, as well as the spatial correlation of the jumping waters.

Figure 1: (a) Distribution of the fraction of defects for the SPC/E water model for different system size. (b) Distribution of the fraction of molecules performing large angular swings ($\Delta\theta > 60^\circ$) within a time window of 200 fs for different system size.

As anticipated by the reviewer, Figure 1 a and b show the distribution of the number of defects and the fraction of jumps for the SPC/E water model. One clearly sees that the relative fluctuations decrease as a function of the system size. The average concentration of the defects as well as the average fraction of jumping waters are, however, not sensitive to the box size. Figure 2 shows the time series associated with the number of jumps and defects for the mid-sized box consisting of 3400 water molecules, along with the correlation plots which demonstrate that the coupling persists.

While the preceding analysis shows that in the thermodynamic limit the waves in the number of jumps and defects will effectively disappear, what is critical for our collective mechanism is the following: 1) that within a sub-system of the macroscopic system, there is a coupling between the jumps and defects and 2) perhaps more importantly, the spatial correlations between the jumping vs random water molecules are not sensitive to finite size effects.

Figure 2: Correlation between the number of simultaneous large angular swings and the number of defects in the system of 3400 molecules for the SPC/E water model, to be compared to Fig.3 in the manuscript obtained for the system of 1019 waters. The correlation coefficients obtained are (b) 0.785 and (c) 0.623.

In order to address further these issues, we begin by showing in Figure 3 top panel, the evolution of the number of jumps for the largest water box consisting of over 8000 waters where one sees clearly the same types of oscillations over time. In the bottom panel of the same Figure, we show the analysis of the number of jumps and defects for a sub-system of approximately 30 Angstroms that is carved out from the mid-sized box (total size 46.720 Angstroms). Although this analysis is more delicate since one has to carefully take into account particles moving in and out of the sub-system, it is clear that there is an obvious coupling between the jumps and defects on similar timescales as our original simulations.

Figure 3: (a) Time series of the number of simultaneous large angular swings in a box of 8152 molecules. (b) Time series of defects and number of jumps of a box subsystem of length 31.1970 Angstroms constructed from the box with 3400 waters (total length 46.720 Angstroms).

Another important check of the consistency and validity of our analysis is how the spatial correlations between the jumping vs random waters (Figure 5 of our current submitted paper) change as a function of finite size. Figures 4 and 5 below compare the spatial correlations between the four nearest jumping versus random water molecules for the three water boxes. It is indeed very comforting and convincing to see that all the spatial distributions essentially overlap with each other for all three system sizes.

We have added the following text and analysis to the manuscript that discusses the finite size sensitivity of our results which includes also new analysis/Figures that are now in the Supplementary Information.

The following sentences have now been added to the Abstract, Introduction, and Results and Discussion, reflecting the sensitivity analysis we perform in response to the three Reviewers including both finite size and water models.

Abstract:

The role of finite size effects, as well as of the chosen water model, on the collective reorientation is also elucidated.

Introduction:

By conducting a finite-size analysis, as well as comparing with several water models, we also elucidate the generality of the collective reorientational mechanism. Specifically, while small-box simulations consisting of approximately 1000 waters, and commonly used in previous studies, tend to enhance the relative fluctuations of the H-bond network underlying the angular jumps, the extent of the spatially correlated clusters of the

Figure 4: Spatial correlations between concurrently jumping molecules for SPC/E water model for systems of different size inspected through the distributions of distances from a jumping water to the first 4 nearest jumping molecules (full lines) compared with those to the first 4 nearest randomly chosen molecules (dashed lines). Jumping molecules are considered to be those that perform swings larger than 60° within 200 fs time interval.

molecules involved, is not affected. Furthermore, the collective reorientational dynamics is found in several water models we examine, with subtle differences in the fraction of defects and asymmetries in the hydrogen-bonding structure both of which can affect the microscopic mechanisms.

Results and Discussion:

One might wonder how much the results we find are affected by the size of the system we consider. In order to discard spurious finite size effects on the collective reorientation mechanism proposed here, we conducted a series of additional simulations for larger system sizes and analyzed their large swings and defect statistics (for details see Methods section and SI, Figures S6-S9). We find that, while the relative magnitude of the large swings and defects fluctuations decreases with the system size, effectively disappearing in the thermodynamic limit, the picture of the reorientation mechanism still persists even for larger systems. Indeed, what is critical for the collective reorientational dynamics is that, also for the large systems, the number of defects and large swings occurring locally in the system are still correlated, and that the spatial correlations between the jumping and the random molecules are not sensitive to finite size effects.

In addition to the lack of a sufficient discussion of finite size effects, I have several comments that are listed below and should be addressed prior to publication:

1. The abstract advertises this work as an example of “unsupervised learning”, which seems to attempt to use language from the machine learning community as a buzz word. However, what is presented is a deterministic analysis of atomistic molecular dynamics simulations and does not involve any “learning” algorithm. Hence, the

Figure 5: Spatial correlations between concurrently jumping molecules for SPC/E water model for systems of different size inspected through the distributions of distances from a jumping water to the first 4 nearest jumping molecules (full lines) compared with those to the first 4 nearest randomly chosen molecules (dashed lines). Jumping molecules are considered to be those that perform swings larger than 40° within 200 fs time interval.

term “unsupervised learning” should be removed because it is misleading potential readers.

Response:

We thank the reviewer for this observation and agree with the overall thrust of the comment. We have now replaced the term *unsupervised learning* with **automatized detection**.

2. *The use of low-pass and smoothing filters (what kind of smoothing filter? Gaussian? Blackman-Harris?) in the frequency domain is explained for the analysis of time traces of the molecular dipole moments and HH-vectors. However, filters are evidently also used to post-process time traces of defect fractions in Figs. 3, S1, S2, S7. Comparisons between Fig. 3 and S1 indicate that different filters are used in different plots of the same property, which is highly confusing especially since the filters aren’t even mentioned in this context.*

Response:

We thank the referee for these questions which we clarify here, as well as in the revised manuscript. The following text has now been added to the manuscript.

On the time traces of the molecular dipole moments and HH-vectors, we first applied a 2nd order low-pass digital butterworth filter (MATLAB implemented, designed to have a frequency response as flat as possible in

the passband). The trajectories are sampled every 4 fs, while the cutoff frequency (the frequency at which the magnitude response of the filter is $1/\sqrt{2}$) which corresponds to 10 THz (1/100 in inverse fs). In order to further smoothen the trajectories before applying the automatized detection of the angular swings, we applied a mean filter using a MATLAB function *smooth* with the span of 25 data points (corresponding to the time window of 100 fs). Furthermore, during post-processing of the time series of defect fractions and number of swings, we have additionally applied a low-pass butterworth filter with the cutoff frequency of 0.1 THz (1/10000 in inverse fs), which allows us to clearly capture the fluctuations on the 10 ps scale. The same smoothing procedures were applied for all the water models and system sizes.

3. To define defects in the hydrogen bond network, a geometrical or energetic criterion for the definition of a hydrogen bond has been used, which is not described in the manuscript.

Response:

We thank the reviewer for carefully identifying this important piece of information that was missing. Indeed for identifying defects and examining which angular swings lead to the breakage of hydrogen bonds or not, we use a geometrical criterion proposed by Luzar and Chandler. This has now been specified in the manuscript with the following text:

To determine the hydrogen bonds, a geometrical criterion is used developed originally by Luzar and Chandler[1]. A pair of water molecules are considered to be hydrogen bonded when the distance between the donating and accepting oxygen atoms (O_D and O_A) is less than or equal to 3.5 Å and the angle formed by the bond vector between the donating hydrogen and oxygen (H_D and O_D) and that between O_D and O_A is less than 30°.

4. I found Figure 5 highly confusing. It seems very strange that distance distributions are shown for the closest two water molecules for large orientational swings, while the distance distributions of the 3rd and 4th closest water molecules are shown for smaller swings. This is very difficult to follow and the reason for this choice is unclear. Fig. S4 shows the complete data set to some degree but the cumulative probability plots are not very useful in this case, since by definition all probabilities add up to 1 and the details of the distribution functions are obfuscated. I recommend to replace Fig. 5 with a version of Fig. S4 that shows the distribution functions instead of cumulative probabilities.

Response:

We thank the reviewer for this observation, we agree that further clarification was needed regarding this point. We have now revised our manuscript to clarify our protocol and justify the angular thresholds employed in Figure 5. Our automated protocol identifies a large spread of swing magnitudes. When a large jump occurs (let's say defined as something that is greater than 60 degrees), nearby water molecules can undergo jumps that are maybe larger or smaller. Our finding was that the differences in the spatial correlations between the random and jumping water molecules for the 3rd and 4th nearest neighbours are more pronounced for a smaller threshold angle of 40 degrees. We thus decided to show those results in the main text. We have now included in the manuscript the following text to clarify this point:

The collective re-orientational motions that we elucidate in our work, involves a spectrum of transitions of both the timescales and angular magnitude, as seen in Figure 4A. Thus, when angular jumps occur in a correlated fashion they will naturally involve a combination of jumps of different magnitude. Although jumping and random spatial distributions for the 3rd and 4th nearest neighbors exhibit dissimilarities also for the jumps with larger angle threshold of 60°, the differences are even more pronounced for the lower angle threshold of 40°.

We have also removed the cumulative distributions from the SI as suggested by the reviewer.

5. *What further puzzles me is the smoothness of the distribution functions in Figure 5. Given that the simulation only contains 1000 water molecules and the statistics are obtained for just 2 nanoseconds, distance distributions between only water molecules that perform large orientational swings within the same 200fs time window should have some noticeable statistical limitations. What bin size was used to compute the distribution functions? Was any smoothing operation applied to the shown curves? (such details were apparently omitted for other plots in the manuscript, as mentioned above, which is why I suspect them here)*

Response:

We thank the reviewer for this question. It should be clarified that although these spatial distributions are constructed from simulations consisting of 1019 water molecules and short simulations of 2 ns, there are a total of 88123920 swings identified out of which 6637487 and 1226684 of them are detected as swings larger than 60 and 40 degrees, respectively. We thus obtain a lot of statistics of these events from our analysis. Nonetheless, the distributions shown in Figure 5 are constructed using a kernel histogram and the kernel width used for constructing the smoothed distributions ranges between 0.1 and 0.2. This information has now been added into the captions of the relevant figures.

Minor comments:

6. *Abstract: ThZ should be THz*

7. *Page 7, paragraph 2, first word: “Armored” should be “Armed”. In any case, a less colloquial formulation may be more suitable*

Response:

We thank the reviewer for identifying the typo, we have, however, omitted using “Armed” in the text.

Reviewer 2 (Remarks to the Author):

I have read the article entitled, "THE COLLECTIVE BURST MECHANISM OF ANGULAR JUMPS IN LIQUID WATER" with great interest. It is a careful study regarding the how the hydrogen bond network reorients using a ML model that relates large reorientations to collective motions. In some sense, it is not surprising that large reorientations require collective motions. Nevertheless quantifying these "bursts" is interesting and important. This study puts a number to this and this is somewhat interesting.

I like Figure 7.

Response:

We thank the reviewer for the positive evaluation of our work.

I think the authors have done an amazing job at quantifying and showing correlations between large rotational swings and the underlying collective structure. They do talk about 2 different water models, but fail to really describe if there are quantitative or qualitative differences between the two models. Obviously, an article in the Nature class journals needs to teach the reader something new about water. I am not sure in its present form this article passes this standard. Don't get me wrong, this paper should be published and will likely be highly cited as the water community is large and diverse in its interests.

I would have liked to see how these jumps change as a description of the local structure—MB-pol vs SPC/E even DFT-based. The class of models such as SPC/E are classified as "frustrated charge" models. They do describe bulk aspects reasonably well (pretty good even). This paper is about details, and there are some important details to examine. To the extent MB-Pol is as charge frustrated as SPC/E is not clear. DFT based models have very "smooth" interactions and do not suffer from the rickety nature of the point charge models. Having an understanding about how this influence jump dynamics would be interesting and teach us something in lieu of an experiment. This is the classic problem of short-range vs long-range and how they are coupled. There is an opportunity to teach us about the nature of this coupling. Do all water models exhibit the same "burst" dynamics?

I would urge the authors to address these issues if they want to be in the Nature class journals. Otherwise this is a great technical study that should be accepted immediately in trade journal such as JCP or JPC.

Response:

We thank the reviewer for all these comments. The question of the sensitivity of the collective reorientational mechanism to the choice of the water model is also in line with some of the concerns posed by Reviewer 3. In order to address these points, we have done a series of systematic checks comparing our results to the TIP5P and the flexible SPC/E water models.

The choice of comparing with the TIP5P model was inspired by our reading of a paper by Remsing and co-workers (J. Phys. Chem. Lett. 2014, 5, 16, 2767–2774)[2] where the authors demonstrate that the electrostatic potential in the center of a cavity is rather sensitive to the choice of water model. Specifically, they showed that three point charge models induce a bigger asymmetry in this potential on donating vs accepting side of hydrogen bonds compared to the five-site models (TIP5P for example). TIP5P appears to be in close agreement with the DFT based simulations at least for neutral cavities. We thus focused our efforts on comparing our results from SPC/E to TIP5P allowing for the simulation of larger boxes and long simulation times. These two water models also provide nice limiting cases within the point-charge construct, to examine how the description of the lone-pair electrons affects the reorientational dynamics.

We begin by showing the time-series evolution of both the number of defects and swings identified by our protocol for the TIP5P simulations in Figure 6. Also shown are the density plots illustrating the correlation between the jumps and defects. Comparing SPC/E and TIP5P, the correlation coefficients are 0.5604 and 0.4596, respectively, for the jumps

Figure 6: Correlation between the number of simultaneous large angular swings and the number of defects in the system of 1019 water molecules for the TIP5P water model, to be compared to Fig.3 in the manuscript obtained for the SPC/E water model. The correlation coefficients obtained are (b) 0.56 and (c) 0.4596.

larger than 60 degrees. Thus, from a birds-eye-view, it appears as though that the burst mechanics of the angular jumps are consistently found in both the 3-site and 5-site water models. The angular jumps in TIP5P also exhibit similar spatial correlations as probed by comparing the distribution of the distances from a jumping water to its 1st, 2nd, 3rd and 4th nearest jumping waters vs first 4 randomly chosen water molecules (see Figures 7 and 8). In these figures, the results for the SPC/E are also overlapped with the TIP5P results. The findings are all self-consistent, namely that jumping water molecules tend to be closer to each other than random ones in the network.

In order to dissect better and understand the differences between the collective jump mechanism obtained with SPC/E and TIP5P, we compared the distribution of the fraction of large jumps for the SPC/E and TIP5P water models which is shown in Figure 9. This suggests that TIP5P incurs a larger number of swings which is also reflected in the larger fraction of defects – specifically, the TIP5P water model increases the average percentage of defects from 48 % to 59 %. By examining and comparing the distribution of different types of defects for SPCE and TIP5P (Figure 10 left and right panel respectively), we observe that the latter enhances the concentration of the 1in-2out, 2in-1out and also the 1in-1out defect water molecules. This is not too surprising since the presence of the additional sites in the TIP5P model analogous to the lone-pair Wannier centers, in DFT enthalpically stabilizes hydrogen bonds that would otherwise be less stable in a 3-site model. This enhancement of the concentration of the defects serves as a seed for observing a larger number of swings in the TIP5P model.

Another aspect of the reorientational mechanism that is likely to reflect more differences between the water models is the asymmetry in the hydrogen bond breaking that occurs as a function of the magnitude of the swing. Figure 11 compares these characteristics for our original SPC/E simulations and the TIP5P water model. Interestingly, we observe that in TIP5P, for large swings, a fewer fraction of them lead to the breakage of hydrogen bonds compared to SPCE. For example, for an angular swing of 60 degrees in the SPCE model, swings detected via the dipole and HH vector involve 85% and 75% hydrogen bond breaking events, respectively. These numbers should be contrasted to 65% and 60% for the TIP5P water model.

Besides the preceding observations, Figure 11 also shows the extent to which swings result in breakage of hydrogen bonds on both the donating (which is naturally expected) and accepting side. This feature shows that angular jumps often come in pairs of water molecules and is another indirect evidence of the collective nature of the reorientational dynamics. Comparing TIP5P and SPC/E we observe that in the former, this feature is enhanced by roughly 10-20%

Figure 7: Spatial correlations between concurrently jumping molecules for TIP5P water model inspected through the probability distribution functions of the distances from a jumping water to the first 4 nearest jumping molecules (full lines) compared with those to the first 4 nearest randomly chosen molecules (dashed lines). Jumping molecules are considered to be those that perform swings larger than 60° within 200 fs time interval.

and is once again consistent with the enhanced angular swings observed in TIP5P.

In conclusion, comparing the SPC/E and TIP5P water models we observe the following:

- The collective nature of the reorientational dynamics is found in both water models.
- Angular swings are enhanced in the 5-site model compared to the 3-site model. This is consistent with the enhanced fraction of defects found in TIP5P vs SPCE.
- Angular swings in TIP5P appear to result in a fewer fraction of hydrogen-bond breaking events compared to SPCE. However, in the case of TIP5P these tend to involve a large number of events where water molecules donating hydrogen bonds to the swinging water, also undergo swings.

We have included in the manuscript a discussion comparing the similarities and differences between the collective angular jump reorientational mechanisms between the SPCE and TIP5P water models. Reviewer 3 also asked for how sensitive our results are to using the flexible SPCE water model.

Although the system sizes are significantly smaller, we also comment on the trends of the re-orientational jumps coming in waves obtained using the MB-pol water model (see SI Figure S20). These results are summarized in the responses that follow and are also included in the manuscript.

Figure 8: Spatial correlations between concurrently jumping molecules for TIP5P water model inspected through the probability distribution functions of the distances from a jumping water to the first 4 nearest jumping molecules (full lines) compared with those to the first 4 nearest randomly chosen molecules (dashed lines). Jumping molecules are considered to be those that perform swings larger than 40° within 200 fs time interval.

Figure 9: Comparison for the SPC/E and TIP5P water model for a system of 1019 molecules. (a) Distribution of the fraction of jumping water molecules with $\Delta\theta > 60^\circ$ within 200 fs. (b) Distribution of the fraction of defects.

Figure 10: Distribution of local defects for the SPC/E and TIP5P water model for a system of 1019 molecules. The axis in the plane depict number of ingoing and outgoing hydrogen bonds.

Figure 11: Fraction of swings of magnitude greater than certain angle threshold $\Delta\Theta$ that break bonds for the TIP5P (full lines) and the SPC/E (dashed lines).

For the SPC/E and TIP5P comparison, we added the following text and a new figure to the Results and Discussion section of the manuscript:

In order to examine the sensitivity of our collective orientational mechanism to the choice of the water model, we compared the rigid SPC/E to both the flexible version of SPC/E[3] and the TIP5P water model[4]. The flexible version of SPC/E has been shown to bring many of the structural and dynamical properties of liquid water into closer agreement with the experiments. Remsing and co-workers recently demonstrated the electrostatic potential in the center of a cavity is rather sensitive to the choice of water model. Specifically, they showed that three-point-charge models induce a bigger asymmetry in this potential on donating vs accepting side of hydrogen bonds compared to the five-site model TIP5P that appears to be in close agreement with the DFT based simulations[2]. Comparing SPC/E and TIP5P water models in terms of the mechanisms we

introduce in this work, provides a way to probe limiting cases of how the lone-pair electrons are treated within point charge water models[5]. We thus focused our efforts on comparing our results from SPC/E to TIP5P circumventing the challenges of using DFT based *ab initio* molecular dynamics. Although we were limited to small system sizes, we also find using the many-body polarizable water model, MB-pol[6, 7], that the jumps also come in waves (see SI Figure S20).

Figure 12: Collective burst mechanism for different point-charge water models. (A,B) Density plots of fraction of defects in the H-bond network versus the number of large jumps within 1ps for TIP5P water model and SPC/E flexible water model, respectively. The correlation coefficients found are (A) 0.4596 and (B) 0.50. (C) Probability distribution function of the fraction of defects for different water models. (D) Fraction of H-bonds that break on the donating vs accepting side, as a function of the magnitude of angular swing for SPC/E and TIP5P water models.

Our analysis shows that for both the TIP5P and flexible SPC/E water models, one observes a correlation between the number of jumps and defects as seen in panels A and B of Fig.12. Similarly, both water models also exhibit the same trends for the differences between spatial correlations observed for the jumping versus random water molecules (Figures S11-S12, S18-S19 in the SI). In the case of TIP5P, jumping water molecules tend to be found clustered slightly closer together in space compared to SPC/E, while for the flexible version of SPC/E, the effect is reversed, with jumping waters clustered further away compared to what is found in the rigid model.

To understand better how the different water models affect the collective reorientational mechanism, we also compared the distribution of the number of defects for the three water models which is shown in Fig.12C. Interestingly, the fraction of defects is TIP5P is enhanced relative to SPC/E while in the flexible version of the SPC/E model, the concentration of defects is reduced. For TIP5P, we observe an enhancement in the concentration of the 1in-2out, 2in-1out and also the 1in-1out defect water molecules (Fig.S14 in SI). This is not

too surprising since the presence of the additional sites in the TIP5P model, analogous to the lone-pair Wannier centers in DFT, enthalpically stabilizes hydrogen bonds that would otherwise be less stable in a 3-site model. This enhancement of the concentration of the defects serves as a seed for observing a larger number of swings in the TIP5P model.

The enhancement of the collective reorientations in the TIP5P water model can also be observed in the fraction of hydrogen bonds that break during swing events. Fig.12D compares the asymmetry in the number of hydrogen bonds that break on the donating vs accepting side, as a function of the magnitude of the angular swing for the SPC/E and TIP5P water model. In the TIP5P water model, there is a larger number of swings where water molecules on the donating side also involve hydrogen bond breaking events. In other words, there are more pairs of water molecules that undergo jumps.

For the case of flexible SPC/E, the fraction of defects decreases relative to the rigid model, see Fig.12C (also see Fig.S17 in the SI). In the flexible water model, the O-H bonds being more delocalized enhances the strength of the hydrogen bonds and therefore reduces the concentration of defects from approximately 48% to 37%. Due to this difference, jumping (and therefore defective) water molecules are spatially correlated farther away from each other than in the rigid model.

Reviewer 3 (Remarks to the Author):

This is an interesting computational work aimed at understanding if angular jumps in water are correlated or not.

As the authors report, the goal of this work is definitely very high. The authors describe their approach quite nicely, and the article is well written, but I am not quite convinced about the soundness of the results.

Response:

We thank the reviewer for the positive evaluation of our work.

In the following, I list the major points I would like the authors to address:

1. The authors performed simulations with 1019 water molecules described by the SPC-E model. It would be useful to show that the results are not affected by the system size and that they are general for various classical potentials. In the SI the authors report data for MB-POL, but not for 512 molecules. Therefore, I would not take them into account. While I understand the request of consistency in using the same model as in the original paper by Laage and Hynes, the SPC-E model is very poor. More realistic models need to be considered. In particular, it would be important to be sure that flexible models show similar jumps and analyse the existence of correlations.

We thank the reviewer for this poignant comments. As indicated in the responses to Reviewer 1, we have done an extensive sensitivity analysis of the role of finite size effects on our collective angular jump mechanisms. In addition, in response to Reviewer 2, we also conduct a comparison of the mechanisms obtained using 3-site (SPC/E) and 5-site water models (TIP5P). We refer the reviewer to this analysis documented earlier in our response letter (Reviewer 1 and Reviewer 2).

To address the reviewers concerns and question on the role of flexible water molecules, we repeated our analysis using 1019 water molecules and using the flexible SPC/E water model[3]. We begin by showing in Figure 13 below the time-series and correlation plots obtained for the flexible SPC/E water model. It is indeed comforting to see that most of the essential features are reproduced in this model. The correlation coefficient of the number of defects and angular swings is 0.5 which is to be compared with 0.56 obtained for the rigid-body SPC/E water model (bottom right panel of Fig.3 in our submitted manuscript).

The most significant difference between flexible and rigid-body SPC/E is the fraction of the number of defects. We find that the flexible water model reduces the fraction of defects by roughly 10%. This feature is consistent with previous reports showing that flexible water molecule enhances the strength of hydrogen bonds in the bulk and also increases the surface tension at the air-water interface[3] bringing both the static and dynamics properties into closer agreement with experiments.

We also compared the extent of the spatial correlations between the jumping and random waters using the flexible SPC/E (Figure 4 and Figure 5). The differences between the random vs jumping waters persists for the flexible model and in fact, the reduced concentration of defects in this model leads to the spatial correlations that are farther away from each other compared to the rigid model.

A discussion on the comparison of all the water models has now been added to the manuscript and can be found in the Response to Reviewer 2.

2. Fig. 5 should be complemented with the RDF computed between ‘jumping’ molecules and compared to the random distribution. This is because the RDF for a random distribution is known, while the distributions reported in the current fig. 5 for random molecules are somewhat obscure.

Figure 13: Correlation between the number of simultaneous large angular swings and the number of defects in the system of 1019 water molecules for the SPC/E flexible water model, to be compared to Fig.3 in the manuscript obtained for the SPC/E rigid water model. The correlation coefficients obtained are (b) 0.61 and (c) 0.50

Figure 14: Comparison for the SPC/E rigid and SPC/E flexible water model for a system of 1019 molecules. (a) Distribution of the fraction of jumping water molecules with $\Delta\theta > 60^\circ$ within 200 fs. (b) Distribution of the fraction of defects.

The purpose of this analysis is to identify possible differences between the *proximal* spatial distributions of water molecules that are simultaneous jumpers versus those that are **randomly distributed** with respect to a jumping water molecule. Although we agree with the reviewer that the random distribution for the bulk water is known, it involves the pairwise correlations between all pairs of water molecules as a function of distance. The distributions we show however, are those associated with the *sorted pairwise distances* between water molecules yielding the 1st, 2nd, 3rd and so-forth nearest neighbours. The RDFs for these distributions are equally obscure in this regard but serve as an important reference to compare with. We have thus left these as is in the manuscript and now provide statistical tests (Kolmogorov-Smirnov[8]) that reinforce the statistical significance of the differences in the distributions we show. The Kolmogorov-Smirnov tests reject the null-hypothesis that both distributions are sampled from the same

Figure 15: Spatial correlations between concurrently jumping molecules for SPC/E flexible water model inspected through the probability distribution functions of the distances from a jumping water to the first 4 nearest jumping molecules (full lines) compared with those to the first 4 nearest randomly chosen molecules (dashed lines). Jumping molecules are considered to be those that perform swings larger than 60° within 200 fs time interval.

distribution with a 5% significance level with p-value significantly smaller than 0.01.

3. Fig. 3A shows, even by eye, some appealing correlation. On the other hand, the same plot reported for MB-pol in the SI shows much less correlation, suggesting that polarisation and/or system size might play a crucial role.

We agree with the reviewer that the MB-pol simulations should be taken with a grain of salt given the small system sizes. As indicated earlier, we now focus the comparison of our results of the SPC/E water model examining finite size effects as well as comparing how the flexible SPC/E and TIP5P water models alter the mechanisms.

Minor points:

1. The asymmetry between donors and acceptors in liquid water is quite known and well described in the literature. The authors should acknowledge previous works with appropriate citations (*J. Chem. Phys.* 141, 084502 (2014), *J. Mol. Liq.* 329, 115530 (2021), *PNAS Nexus* 1, 1-8 (2022)) and frame their results based on this previous knowledge.

Response:

We thank the reviewer for pointing us to this literature. We have now added these references and discussed our results in the context of these previous studies (see the following highlighted text). We believe that many of the points

Figure 16: Spatial correlations between concurrently jumping molecules for SPC/E flexible water model inspected through the probability distribution functions of the distances from a jumping water to the first 4 nearest jumping molecules (full lines) compared with those to the first 4 nearest randomly chosen molecules (dashed lines). Jumping molecules are considered to be those that perform swings larger than 40° within 200 fs time interval.

discussed in these papers are best placed in our current narrative, within the discussion and conclusions.

Our findings on the collective nature of angular jumps in bulk water have important implications and connections with previous studies inspecting the properties of water across the phase diagram. In the recent years, Martelli and co-workers have quantified the complexity and topology of the hydrogen-bond network in terms of both ring-statistics[9] and topological defects akin to our analysis, both at room temperature and under supercooled conditions[10]. In addition, Di Stasio and co-workers have studied using *ab initio* molecular dynamics the role of electronic structure in altering the relative population of topological defects in bulk water[11]. A natural extension of our work, would be to explore how the collective reorientational dynamics changes under supercooled conditions although this would require significantly longer simulations to converge. This may be an ideal playground for the use of deep neural networks[12] as well as many-body polarizable potentials[6, 7] where the role of electronic effects and polarization on the reorientational mechanism can be investigated in the future.

2. Fig. 6: could be related to the decoupling of translational and rotational degrees of freedom occurring in the supercooled regime, as hypothesised in PNAS Nexus 1, 1-8 (2022). The authors should mention this.

Response:

We thank the reviewer for this suggestion which is indeed very interesting. In our discussion centered around Figure 6 in the manuscript, we now add a sentence making this connection with the addition of the following sentence:

The apparent relationship between the magnitude of angular jumps and the local-density fluctuations suggests a possible coupling between the fluctuations of the network involving translational and rotational degrees of freedom. It would be interesting in future studies to examine this effect in supercooled water as recently highlighted by Martelli[10].

3. *The use of the word ‘topology’ in this manuscript is out of context, as the authors do not provide any analysis about the topology of the hydrogen bond network.*

Response:

We thank the reviewer for this careful reading of the language we use. We have now used the terms “local topology” or “local coordination topology” when we needed to refer to the changes in the connections between the water molecules of the H-bond network due to the breakage and formation of H-bonds.

4. *The authors refer to “Automated detection and unsupervised detection”. Such wording infer the presence of a ML algorithm but it is not to be the case. The wording should be modified accordingly.*

Response:

This point was also raised by Reviewer 1. We have eliminated from the manuscript any reference to *unsupervised detection*. However, our protocol does provide an automatic manner for identifying and quantifying the collective nature of angular swings and so we keep the language of **automatic detection**.

5. *In the SI it is reported that large angular jumps occur in the order of 0.1 ps. On the other hand, density fluctuations occur on the order of 4ps (N.J. English and J.S. Tse, Phys. Rev. Lett. 106, 037801 2011, also cited by the authors), on a time scale comparable to that of hydrogen-bonded-related fluctuations (Santra et al., Mol. Phys. 113, 2829-2841, 2015).*

Response:

The timescale of the large angular jumps that we find in Fig.4A (also in Fig.S4 in the current SI) is indeed of the order of 0.1 ps. Note, however, that this is the timescale found from the automatized detection of large angular jumps and as such reports the timescale over which the angular swing occurs. It does not include the time needed for H-bond breakage and formation, which is indeed on the order of picoseconds.

In conclusion, I am not convinced that the ‘jumps’ are spatially correlated. The authors need to address my points before I can accept the manuscript for publication in nature communications.

Response:

We believe that with the additional results we incorporated into the manuscript we have addressed all the comments of all the Reviewers and significantly improved the quality of the work. We thank them again for their interest and questions.

Other Changes/Edits to the Manuscript:

Finally, besides all the preceding changes, we also include a discussion of relevant previous literature pointing to the collective fluctuations in water. This also includes a connection between our results and dielectric spectroscopy as well as its implications on collective reorientational dynamics near biological systems and under supercooling. The added text is the following:

As seen from the previous analysis on the changes in the local environment involving the local topology of the hydrogen bond network, large swings must naturally result in disruption of the hydrogen bond network of at least one of the nearest neighbors. Furthermore, large angular swings also tend to occur in low density and more disordered environments. Since density fluctuations involve collective reorganizational processes within the hydrogen bond network, large swings could presumably either lead to consequent angular jumps of the near by molecules, as suggested in the literature [13, 14, 15] or alternatively, be part of several large angular swings that occur simultaneously.

In addition, the dielectric relaxation of water has been shown in several studies to involve the collective relaxation of water dipoles[16, 17]. The non-local reorientation mechanism we elucidate here is consistent with this picture.

Although we have focused on bulk water at room temperature showing the highly cooperative character of the reorientational dynamics of the water molecules, we believe that our results open the doors to exploring how this effect changes upon supercooling and near biological systems where one might expect an enhancement of the phenomenon. Very recent results by Laage and coworkers suggest that the HB-breaking angular events are related to water translational diffusion, pinpointing the connection between water's collective reorientation and its transport properties [18].

References

- [1] Alenka Luzar and David Chandler. Hydrogen-bond kinetics in liquid water. *Nature*, 379(6560):55–57, 1996.
- [2] Richard C Remsing, Marcel D Baer, Gregory K Schenter, Christopher J Mundy, and John D Weeks. The role of broken symmetry in solvation of a spherical cavity in classical and quantum water models. *The journal of physical chemistry letters*, 5(16):2767–2774, 2014.
- [3] Yujie Wu, Harald L. Tepper, and Gregory A. Voth. Flexible simple point-charge water model with improved liquid-state properties. *The Journal of Chemical Physics*, 124(2):024503, 2006.
- [4] Michael W. Mahoney and William L. Jorgensen. A five-site model for liquid water and the reproduction of the density anomaly by rigid, nonpolarizable potential functions. *The Journal of Chemical Physics*, 112(20):8910–8922, 2000.
- [5] Richard C. Remsing, Timothy T. Duignan, Marcel D. Baer, Gregory K. Schenter, Christopher J. Mundy, and John D. Weeks. Water lone pair delocalization in classical and quantum descriptions of the hydration of model ions. *The Journal of Physical Chemistry B*, 122(13):3519–3527, 2018. PMID: 29378124.
- [6] Volodymyr Babin, Claude Leforestier, and Francesco Paesani. Development of a “first principles” water potential with flexible monomers: Dimer potential energy surface, vrt spectrum, and second virial coefficient. *Journal of chemical theory and computation*, 9(12):5395–5403, 2013.
- [7] Sandeep K Reddy, Shelby C Straight, Pushp Bajaj, C Huy Pham, Marc Riera, Daniel R Moberg, Miguel A Morales, Chris Knight, Andreas W Götz, and Francesco Paesani. On the accuracy of the mb-pol many-body potential for water: Interaction energies, vibrational frequencies, and classical thermodynamic and dynamical properties from clusters to liquid water and ice. *The Journal of chemical physics*, 145(19):194504, 2016.
- [8] Marvin Karson. Handbook of methods of applied statistics. volume i: Techniques of computation descriptive methods, and statistical inference. volume ii: Planning of surveys and experiments. i. m. chakravarti, r. g. laha, and j. roy, new york, john wiley; 1967, \$9.00. *Journal of the American Statistical Association*, 63(323):1047–1049, 1968.
- [9] Fausto Martelli. Topology and complexity of the hydrogen bond network in classical models of water. *Journal of Molecular Liquids*, 329:115530, 2021.

- [10] Fausto Martelli. Steady-like topology of the dynamical hydrogen bond network in supercooled water. *PNAS Nexus*, 1(3), June 2022. eprint: <https://academic.oup.com/pnasnexus/article-pdf/1/3/pgac090/44817770/pgac090.pdf>.
- [11] Robert A. DiStasio, Biswajit Santra, Zhaofeng Li, Xifan Wu, and Roberto Car. The individual and collective effects of exact exchange and dispersion interactions on the ab initio structure of liquid water. *The Journal of Chemical Physics*, 141(8):084502, 2014.
- [12] Thomas E. Gartner, Linfeng Zhang, Pablo M. Piaggi, Roberto Car, Athanassios Z. Panagiotopoulos, and Pablo G. Debenedetti. Signatures of a liquid–liquid transition in an ab initio deep neural network model for water. *Proceedings of the National Academy of Sciences*, 117(42):26040–26046, 2020.
- [13] Chao Liu, Wenfei Li, and Wei Wang. Correlation of reorientational jumps of water molecules in bulk water. *Physical Review E*, 87(5):052309, 2013.
- [14] Marco G Mazza, Nicolas Giovambattista, Francis W Starr, and H Eugene Stanley. Relation between rotational and translational dynamic heterogeneities in water. *Physical review letters*, 96(5):057803, 2006.
- [15] Iwao Ohmine, Hideki Tanaka, and Peter G Wolynes. Large local energy fluctuations in water. ii. cooperative motions and fluctuations. *The Journal of chemical physics*, 89(9):5852–5860, 1988.
- [16] Shinji Saito and Iwao Ohmine. Dynamics and relaxation of an intermediate size water cluster (h₂o)₁₀₈. *The Journal of Chemical Physics*, 101(7):6063–6075, 1994.
- [17] Daniel C. Elton. The origin of the debye relaxation in liquid water and fitting the high frequency excess response. *Phys. Chem. Chem. Phys.*, 19:18739–18749, 2017.
- [18] Axel Gomez, Zeke A. Piskulich, Ward H. Thompson, and Damien Laage. Water diffusion proceeds via a hydrogen-bond jump exchange mechanism. *The Journal of Physical Chemistry Letters*, 13(21):4660–4666, 2022. PMID: 35604934.

REVIEWERS' COMMENTS

Reviewer #1 (Remarks to the Author):

The authors have addressed my concerns and significantly improved the original manuscript. A careful analysis of finite size effects was added, which is crucial to support the findings of this study. This was further accompanied by a comparative study with additional water models, which verified the general mechanism.

In addition, the authors clarified and clearly specified the use of filters in their analysis, which is critical for the reproduction of the presented results. Further, the language regarding the learning mechanisms has been adapted in response to comments from me and the other reviewers.

The reported findings of correlated reorientations of water molecules provide a new quality of understanding to collective dynamics in water and aqueous solutions and thus, in my opinion, warrant publication in Nature Communications.

Reviewer #2 (Remarks to the Author):

The authors have performed a careful rebuttal--I have no further issues.